# MenT nucleotidyltransferase toxins extend tRNA acceptor stems and can be inhibited by asymmetrical antitoxin binding

Xibing Xu[1,4], Ben Usher[2,4], Claude Gutierrez [3], Roland Barriot [1], Tom J. Arrowsmith[2], Xue Han[1], Peter Redder [1], Olivier Neyrolles [3], Tim R. Blower [2,5] ✉ & Pierre Genevaux [1,5] ✉

*Mycobacterium tuberculosis*, the bacterium responsible for human tuberculosis, has a genome encoding a remarkably high number of toxin-antitoxin systems of largely unknown function. We have recently shown that the *M. tuberculosis* genome encodes four of a widespread, MenAT family of nucleotidyltransferase toxin-antitoxin systems. In this study we characterize MenAT1, using tRNA sequencing to demonstrate MenT1 tRNA modification activity. MenT1 activity is blocked by MenA1, a short protein antitoxin unrelated to the MenA3 kinase. X-ray crystallographic analysis shows blockage of the conserved MenT fold by asymmetric binding of MenA1 across two MenT1 protomers, forming a heterotrimeric toxin-antitoxin complex. Finally, we also demonstrate tRNA modification by toxin MenT4, indicating conserved activity across the MenT family. Our study highlights variation in tRNA target preferences by MenT toxins, selective use of nucleotide substrates, and diverse modes of MenA antitoxin activity.

Toxin-antitoxin (TA) systems comprise small genetic modules encoding a noxious toxin and its antagonistic antitoxin. They are widespread throughout bacterial and archaeal genomes, and on mobile genetic elements, and are generally stress inducible[1–5]. TA systems have roles in defending against phage infection, the maintenance of genomic regions, and in some cases, contribute to bacterial virulence and antibiotic persistence[4,6–9]. It has been shown that under lenient growth conditions, toxin activity is blocked by its cognate antitoxin and bacterial growth is not impacted. However, under specific conditions such as phage infection or plasmid loss, the balance of toxin and antitoxin is dysregulated. As a result, free toxins target essential cellular processes or structures, including translation, replication, metabolism or the cell envelope, causing growth inhibition or cell death[2,10].

*Mycobacterium tuberculosis*, the bacterium responsible for human tuberculosis, has a genome that encodes a remarkable

abundance of over 86 TA systems[11,12]. This suite of TA systems includes multiple examples from well-conserved TA families, which have generally been shown to be induced under relevant stress conditions, including hypoxia, macrophage engulfment, or exposure to antibiotics[13,14]. Many of the putative *M. tuberculosis* toxins were shown to be toxic when expressed in *M. tuberculosis*, *M. smegmatis* and/or *E. coli*, while their deleterious effect was efficiently inhibited by co-expression of the corresponding antitoxin[13,15]. Accordingly, it has been proposed that activated toxins could modulate *M. tuberculosis* growth under certain conditions, thereby contributing to survival in the human host[11,15,16]. Yet, except for a handful of TA systems that were tested and shown to contribute to host infection[17–19], their cellular function remains largely unknown. In addition, the highly toxic nature of some of these toxins suggests that their antibacterial mechanisms could be used to identify

[1]Laboratoire de Microbiologie et Génétique Moléculaires (LMGM), Centre de Biologie Intégrative (CBI), Université de Toulouse, CNRS, Université Toulouse III - Paul Sabatier (UT3), Toulouse, France. [2]Department of Biosciences, Durham University, South Road, Durham DH1 3LE, UK. [3]Institut de Pharmacologie et de Biologie Structurale (IPBS), Université de Toulouse, CNRS, Université Toulouse III - Paul Sabatier (UT3), Toulouse, France. [4]These authors contributed equally: Xibing Xu, Ben Usher. [5]These authors jointly supervised this work: Tim R. Blower, Pierre Genevaux. ✉e-mail: timothy.blower@durham.ac.uk; pierre.genevaux@univ-tlse3.fr

new drug targets, or alternatively, via direct application as intracellular antimicrobials[20–24].

Members of the MenAT family of TA systems in *M. tuberculosis* encode a toxin with a conserved nucleotidyltransferase (NTase)-like protein domain (DUF1814) and a cognate antitoxin, which can belong to different protein families[25,26]. Among the MenAT family members of *M. tuberculosis*, MenT1 (Rv0078A), MenT3 (Rv1045) and MenT4 (Rv2826c) were shown to inhibit *M. smegmatis* growth when over-expressed, while MenT2 (Rv0836c) showed no detectable effect[25]. MenT toxins belong to the family of abortive infection proteins (AbiEii) of *Streptococcus agalactiae* that contain four conserved signature motifs[26]. The N-terminal motifs I and II are found in DNA polymerase β and are proposed to co-ordinate a metal ion for nucleotide binding and transfer. The C-terminal motif III is similar to that of tRNA NTases that add the 3′ CCA motif to immature tRNAs, and may be important for base stacking with substrates. The C-terminal motif IV is unique to DUF1814 proteins and is proposed to form a catalytic site with motif III[26,27]. To date, the MenT3 toxin is the best-characterized member of this family. MenT3 (also named TglT) was shown to inhibit translation by transferring pyrimidines to the 3′ CCA end of tRNA acceptor stems (preferentially to Ser-tRNA in vitro), thus preventing further aminoacylation[25]. Remarkably, such activity was suggested to be neutralized by the MenA3 antitoxin acting as a specific kinase phosphorylating MenT3 at the Ser78 catalytic site residue[27]. This proposed mode of inhibition led to the classification of MenAT3 as a type VII TA system[3,28]. The X-ray structures of MenT3 and MenT4 show that both are monomeric, bi-lobed globular proteins, with similarity in their overall fold, especially in their putative active site[25]. In addition, the structures of antitoxin MenA4 and close homologue AbiEi reveal the presence of N-terminal winged helix-turn-helix DNA-binding domains connected by a short linker to C-terminal kinase domains involved in toxin neutralization[29–31]. In vivo, *menAT2* was shown to be induced upon exposure to nitrosative stress and required for *M. tuberculosis* pathogenesis in guinea pigs[32]. In contrast, virtually nothing is known about MenAT1. The MenAT1 system encodes the expected NTase-like toxin protein MenT1 (sharing 15% amino acid identity with MenT3) together with a very short putative antitoxin of 68 amino acids, named MenA1, originally identified as a putative SymE-like type I toxin and predicted to be disordered and lacking a DNA-binding domain[12]. Previous work showed that MenT1 is toxic when expressed in *M. smegmatis* and that its toxicity was efficiently inhibited by co-overexpressed MenA1[25]. Intriguingly, overexpression of MenT1 did not show detectable toxicity in *E. coli*, which is in sharp contrast with MenT3 or AbiEii[25,33].

In this work, we have investigated the biochemical activity, structure and antitoxicity of MenAT1. We show that MenAT1 can function as a bona fide TA system in vivo in its native host *M. tuberculosis*. Furthermore, the toxin MenT1 acts as a tRNA NTase in vitro, which transfers CMPs to the 3′ end of tRNA acceptor stems without preference for specific tRNAs. MenT1 biochemical activity was fully inhibited by purified MenA1. We present the solved crystal structure of MenT1 alone and in complex with its small protein antitoxin MenA1. The MenA1 monomer prevents MenT1 toxicity by binding asymmetrically to identical faces of two MenT1 protomers, thus blocking the MenT1 active sites. Finally, we demonstrate that MenT4 also shows no strong preference for specific tRNA, indicating a conserved mode of toxicity across the widespread MenT family.

## Results

### MenAT1 is a bona fide TA system in *M. tuberculosis*

Although MenT toxins can be identified by the presence of their DUF1814 NTase domain, MenA antitoxins are more diverse and do not show any detectable crosstalk with non-cognate MenT toxins[25]. Whether such diversity in MenAT pairings reflects different antitoxic mechanisms remains unknown[25,27,29]. MenA1, encoded by *rv0078B*, is a small 68 amino acid protein that does not appear to encode a DNA-binding domain. Within H37Rv, *rv0078B* is found in an "atypical genomic region", suggesting it was potentially acquired by horizontal gene transfer in an ancestor of *M. tuberculosis*[34]. Analysis of the presence of MenA1 homologues in bacterial genomes using FlaGs[35] revealed that out of the 21 *menA1* genes identified, 15 were located upstream of a *menT*-like open reading frame, while the others were unknown genes (Fig. 1A, Fig. S1, Supplementary datasheet 1). Sequence alignments of the identified MenT1 and MenA1 homologues demonstrates conservation of structural elements and putative active site residues (Fig. S2). Although MenAT1 is conserved in all *M. tuberculosis* genomes, nothing is known about its functionality in this bacterium. Therefore, we deleted the *menAT1* operon in *M. tuberculosis* H37Rv and monitored toxicity (expression of *menT1* alone) and toxicity rescue (*menAT1* operon) in both the deletion mutant and wild-type (WT) strains (Fig. 1B). The results show that MenT1 is indeed capable of efficiently inhibiting *M. tuberculosis* growth and that such a deleterious effect was efficiently neutralized by *menA1* present either on the chromosome (H37Rv WT) or on an integrative plasmid (H37Rv Δ*menAT1*). These data show that MenAT1 is a bona fide TA system in *M. tuberculosis*.

### Toxin MenT1 has a conserved nucleotidyltransferase catalytic core

To further investigate MenT1 function, we first determined the X-ray crystallographic structure of MenT1 alone to a resolution of 1.65 Å (Fig. 2A, Table 1, Fig. S3). The MenT1 structure presents a bi-lobed globular protein, with an upper domain consisting of a seven-strand β-sheet decorated either side with two α-helices, connected by a short linker to a second domain made up of a four α-helical bundle (Fig. 2A). Surface electrostatics show stark differences on either face of the protein, with a distinct electropositive cavity on one face (Fig. 2B). We performed sequence-independent structural superpositions of MenT1 using our previously solved structures of toxins MenT3 (PDB: 6Y5U) and MenT4 (PDB: 6Y56). Superposing MenT1 with MenT3 returned a root mean square deviation (RMSD) of 3.829 Å (849 atoms), suggesting a close but not exact alignment, which likely arises from the relatively larger size of MenT3 (Fig. 2C). Nevertheless, the core fold and domain structures are similar (Fig. 2C). Superposition of MenT1 with MenT4 produced an RMSD of 4.232 Å (889 atoms), and a similar alignment of core regions (Fig. 2D). Importantly, when all three solved toxins are aligned, the identified catalytic regions co-localise (Fig. 2E, F). This allowed identification of key conserved catalytic residues for MenT1, namely T39 (S78, S67 in MenT3 and MenT4, respectively), D41 (D80, D69 in MenT3, MenT4), K137 (K189, K171 in MenT3, MenT4), and D152 (D211, D186 in MenT3, MenT4). The four conserved putative catalytic residues of MenT1 identified from the structure were substituted to alanine and tested for their toxicity in vivo when expressed in both *M. tuberculosis* Δ*menAT1* (Fig. 2G) and *M. smegmatis* (Fig. S3E). While D41A, K137A, and D152A substitutions abolished MenT1 toxicity in both bacteria, T39A remained toxic, which is consistent with the toxicity of the corresponding S78A substitution in MenT3. These data show a strong conservation of the proposed catalytic site of MenT-like toxins.

Noticeably, the MenT1 crystal structure has two MenT1 protomers in the asymmetric unit (Fig. S3A). Alignment of both protomers produces a low RMSD of 0.301 Å (1105 atoms), demonstrating that both protomers are very similar (Fig. S3B). The MenT1 crystal structure was submitted to PDBsum[36], to generate a detailed analysis of the respective MenT1 protomer–protomer binding interfaces. While PDBsum calculated a buried surface area of 1134 Å$^2$, the interfaces consisted primarily of only van der Waals interactions. In addition, by size exclusion chromatography (SEC), purified MenT1 was monomeric in solution (Figs. S3C, D). This fits with what is known for MenT3 and MenT4[25] solution states, and confirms that MenT1 is also monomeric,

with the presence of two protomers in the asymmetric unit a result of crystallographic packing.

Structural superposition of MenT1 with ANT2 (PDB 4XJE) demonstrated a potential binding mode for nucleotide substrates, with the triphosphate co-ordinated through divalent metal ions supported by conserved MenT1 residues D41 and D43 (Fig. S4A). Structural superposition of MenT1 with a CCA-adding enzyme from *A. fulgidus* (PDB 3OVA) shows not only a similar potential nucleotide

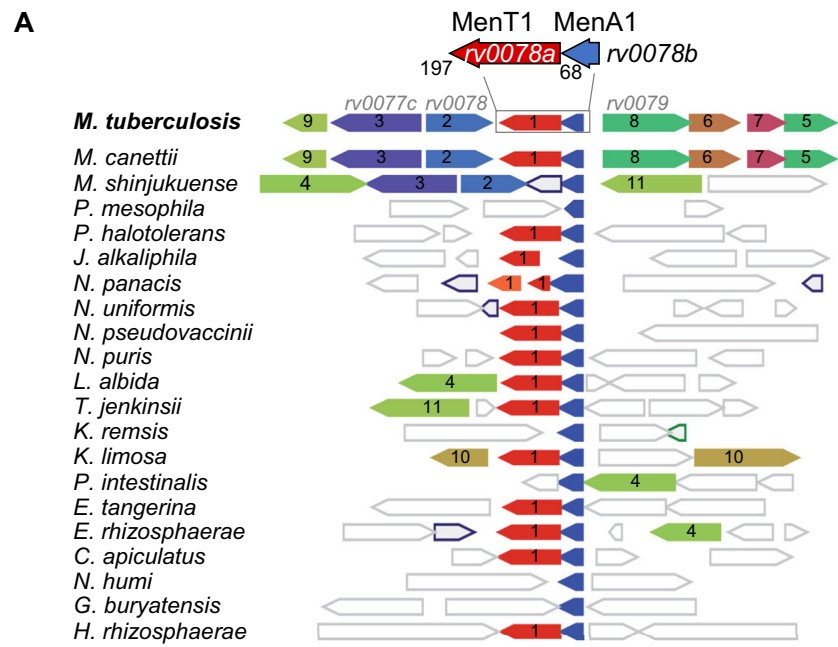

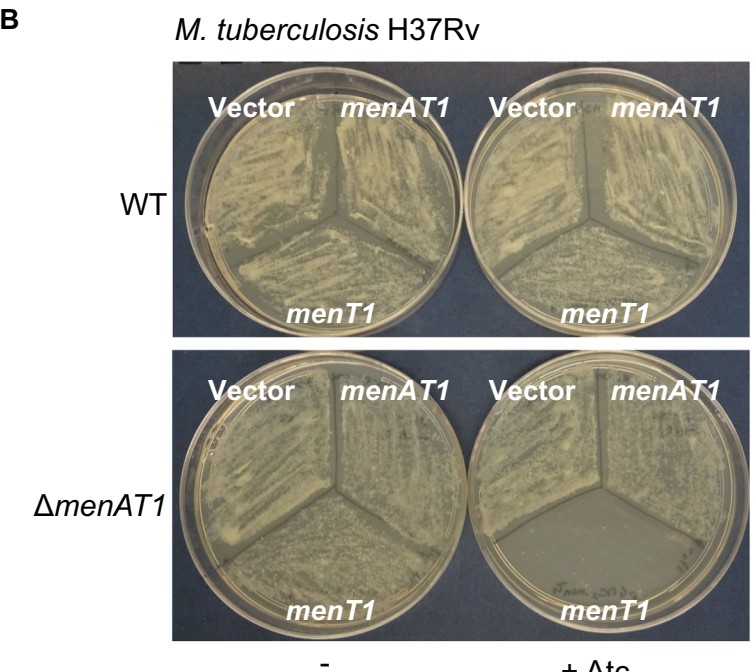

**Fig. 1 | MenAT1 is a bona fide TA system in *M. tuberculosis*. A** Conservation of gene neighbourhoods and maximum likelihood phylogenetic analyses were used to study the distribution of *menA1* genes (performed with FlaGs, http://www.webflags.se/). The *menA1* gene is highlighted in blue, and non-conserved genes are not coloured. The genes encoding proteins that belong to a homologous cluster in more than one genomic neighbourhood are indicated by colour (refer to supplementary datasheet 1 for the identification of clusters and their corresponding flanking protein accession numbers). Among the most conserved adjacent proteins, (2) corresponds to Rv0078 a known repressor of *menTA1*, (3) to Rv0077c a putative oxidoreductase, (8) to Rv0079 unknown gene of the dormancy regulon, (11) to a putative LysR transcription regulator, (4) to a P-loop-NTPase containing domain, and (10) to a rifampin ADP-ribosyltransferase. Full names of bacterial species are given in Fig. S1. **B** *M. tuberculosis* H37Rv or its mutant strain Δ(*menA1-menT1*)::ZeoR was transformed with pGMC vector, pGMC-*menT1*, or pGMC-*menAT1*. Following phenotypic expression, half of the transformation mix was plated on 7H11 OADC plates with Sm, and the other half was plated on 7H11 OADC Sm plates that were supplemented with 200 ng.ml⁻¹ of Atc inducer. The plates were incubated at 37 °C for 20 days. The data presented are representative of three independent experiments.

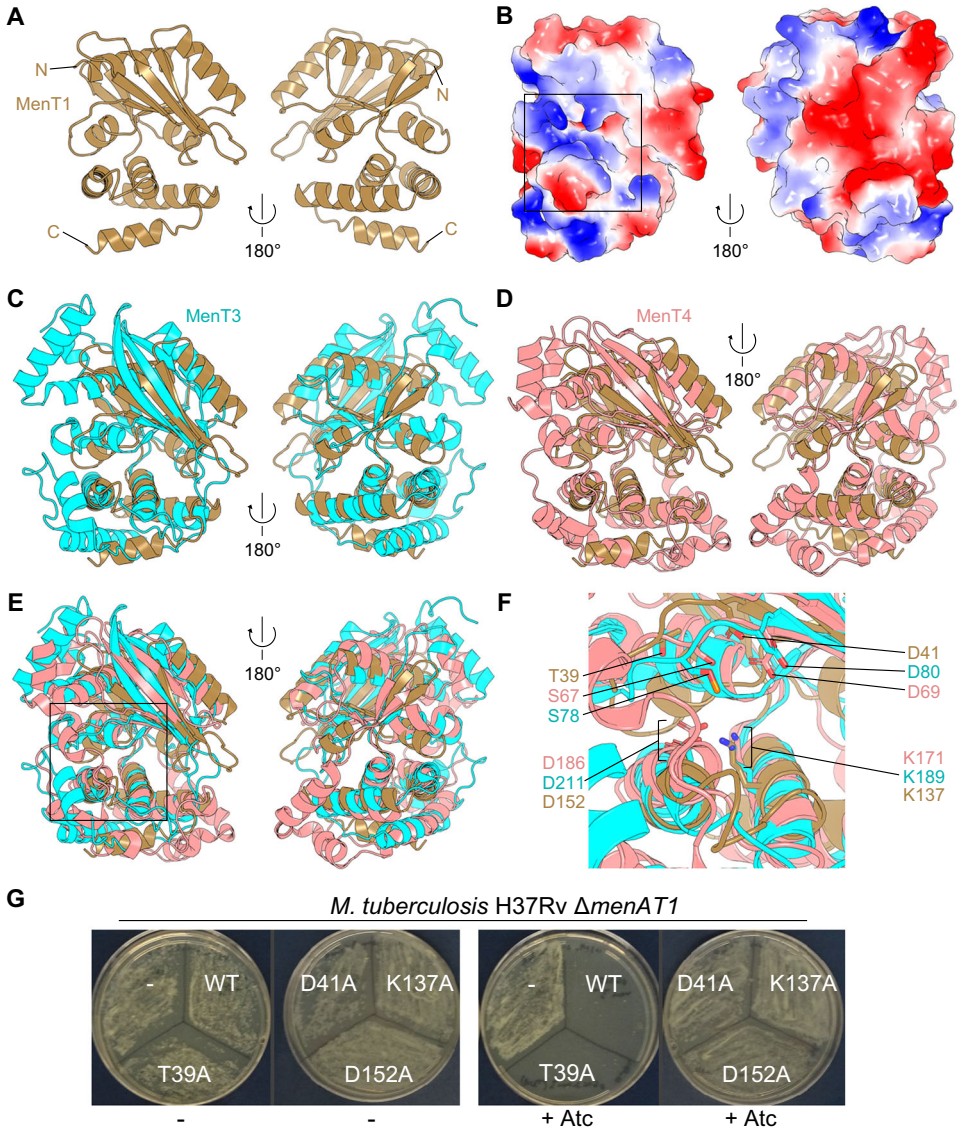

**Fig. 2 | MenT toxins have conserved folds and catalytic sites. A** Structure of monomeric MenT1 toxin, with views from front and back, shown as cartoons coloured "sand". N and C termini are indicated. **B** Surface electrostatics of MenT1, viewed as in (**A**), with red for electronegative and blue for electropositive potential. Electrostatics were generated using default settings for the APBS plugin (PyMol). The box indicates likely site of catalysis. **C** Superposition of MenT1 and MenT3 (cyan cartoon), viewed as per (**A**). **D** Superposition of MenT1 and MenT4 (salmon cartoon), viewed as per (**A**). **E** Superposition of MenT1, MenT3 and MenT4, viewed as per (**A**). **F** Close-up view of boxed region in (**E**), highlighting MenT toxin active site residues. **G** A toxicity assay was performed to evaluate the potential activity centre of MenT1 in *M. tuberculosis*. The mutant strain Δ(*menA1-menT1*)::ZeoR was transformed with 100 ng plasmids expressing either MenT1 WT or mutant alleles T39A, D41A, K137A, or D152A. Following phenotypic expression, half of the transformation mix was plated on 7H11 OADC plates with Sm, and the other half was plated on 7H11 OADC Sm plates that were supplemented with 200 ng.ml⁻¹ of Atc. The plates were incubated at 37 °C for 20 days. The data presented are representative of three independent experiments.

binding mode (Fig. S4B), but also the potential positioning for a tRNA substrate, wrapping over the MenT1 surface and presenting the 3' end at the active site (Fig. S4C).

**MenT1 inhibits protein synthesis and shows tRNA NTase activity**

Previous work showed that MenT3 inhibits protein synthesis in vitro[25]. We purified MenT1 WT and the inactive D41A mutant to test their impact on protein synthesis using the *E. coli* transcription/translation coupled PURE assay, as described previously[25,37]. Initially, multiple attempts with different model substrates did not show any detectable effect of MenT1, which corresponds with the complete lack of MenT1 toxicity in *E. coli*[25]. In contrast with *E. coli*, which possesses several copies of most tRNAs genes[38], both *M. smegmatis* and *M. tuberculosis* only have single copies of most tRNA genes, except for tRNA^Met (CAT)

and tRNA^Cys (GCA)[39], which could make them more sensitive to tRNA modifying toxins. Since the PURE system has an excess of each *E. coli* tRNA, we reasoned that decreasing the tRNA concentration could reveal any MenT1 effects on translation. Accordingly, incubation of MenT1 at lower tRNA concentration demonstrated that MenT1 WT, but not catalytic mutant D41A, could inhibit synthesis of both GFP and WaaF model proteins (Fig. 3A).

The fact that MenT1 is structurally related to MenT3 and also affects protein synthesis suggests that MenT1 could also function as a tRNA NTase. When tested in the presence of labelled [α-³²P]-NTPs, in vitro incubation of MenT1 with either total RNA extracts from *M. smegmatis*, *M. tuberculosis*, and *E. coli*, or purified total tRNA of *E. coli*, showed the appearance of labelled products at the size of tRNAs in all cases, suggesting that MenT1 could indeed transfer nucleotides to

**Table 1 | Crystallographic data collection and refinement statistics**

| | MenT₁ apo native | MenAT₁ complex native |
|---|---|---|
| PDB ID Code | 8AN4 | 8AN5 |
| Number of crystals | 1 | 1 |
| Beamline | Diamond I24 | Diamond I04 |
| Wavelength, Å | 0.9795 | 0.9795 |
| Resolution range, Å | 43.84–1.65 (1.71–1.65)ᵃ | 43.68–1.44 (1.492–1.44) |
| Space group | I 4 2 2 | P 2₁ |
| **Unit cell** | | |
| a b c (Å) | 123.99 123.99 118.34 | 45.87 81.73 64.83 |
| α β γ (°) | 90.00 90.00 90.00 | 90.00 107.78 90.00 |
| Total reflections | 5,659,100 (551,522) | 1,110,358 (107,121) |
| Unique reflections | 55,309 (5434) | 82,208 (8166) |
| Multiplicity | 102.3 | 13.5 |
| Completeness (%) | 90.68 (14.59) | 99.80 (98.70) |
| Mean I/sigma(I) | 8.95 | 13.07 |
| $R_{merge}$ | 1.82 (−20.7) | 0.09 (2.15) |
| $R_{meas}$ | 1.83 (−20.8) | 0.09 (2.24) |
| $CC_{1/2}$ | 0.99 (0.29) | 1.00 (0.63) |
| $R_{work}$ | 0.2247 (0.7550) | 0.1706 (0.3348) |
| $R_{free}$ | 0.2333 (0.9489) | 0.1899 (0.3398) |
| **No. of non-hydrogen** | | |
| Atoms | 3077 | 3679 |
| Macromolecules | 2957 | 3335 |
| Solvent | 120 | 344 |
| Protein residues | 384 | 429 |
| RMSD (bonds, Å) | 0.01 | 0.01 |
| RMSD (angles, °) | 1.41 | 1.12 |
| Ramachandran favoured (%) | 97.88 | 98.56 |
| Ramachandran allowed (%) | 2.12 | 1.44 |
| Ramachandran outliers (%) | 0.00 | 0.00 |
| Average B-factor | 43.87 | 37.70 |
| Macromolecules | 43.97 | 37.42 |
| Solvent | 41.49 | 40.40 |

ᵃStatistics for the highest-resolution shell are shown in parentheses.

tRNA (Fig. 3B). Incubation of MenT1 and total *M. smegmatis* RNA in the presence of each individual labelled nucleotide separately revealed that MenT1 has a strong preference for CTP in vitro, although weaker but detectable modifications could also be observed with the other three tested nucleotides. Noticeably, the D41A mutant was not able to transfer nucleotides (Fig. 3C).

**MenT1 preferentially adds cytidines to tRNA acceptor stems**

The CTP dependence (Fig. 3C) and the multiple bands of labelled tRNAs (Fig. 3B) suggested that MenT1 might add cytidines to different tRNAs. We investigated further by developing and implementing a sequencing method that allows the exact mapping of 3′-OH ends of tRNA (Fig. S5A). Total RNA extracts of *M. smegmatis* (as used in Fig. 3B) were incubated with purified MenT1 in the presence of CTP and the 3′-OH RNA-seq libraries were generated and sequenced (Fig. 3D, E). Our data show that in the absence of toxin, the vast majority (over 99.4%) of the tRNA ends possessed the expected CCA acceptor sequence, whereas in the presence of the toxin ~30% acquired extra cytidines, i.e., 24.3% with CCA + C, 4.9% with CCA + CC and 0.3% with CCA + CCC. When the modification pattern of the 31 individual tRNAs identified

was analysed, we found that all the identified tRNAs were modified to similar levels, which suggested that MenT1 has no apparent preference for specific tRNAs in vitro (Fig. 3E). Note that biological replicates and total reads of all the tRNA seq experiments are given in the Supplementary Datasheet 1. To further confirm that MenT1 can modify most of the tRNA, each of the 45 tRNAs of *M. tuberculosis* were individually transcribed in vitro using T7 RNA polymerase and screened for tRNA modification by MenT1 in the presence of [α-³²P]-CTP. Such a non-quantitative screen, which only detects 3′ modifications of tRNA, confirmed that almost all tRNA, 42 out of the 45 in this case, could be labelled with CMP in the presence of MenT1 (Fig. 4A). Since the 3′ ends generated following T7 transcription are highly heterogeneous[40], we next selected five representative tRNAs from *M. tuberculosis*, namely tRNA-Gly-3, -His, -Leu-3, -Met-2, and -Ser-4 independently labelled them with [α-³²P]-CTP, and purified them using a cleavage method that generated homogeneous 3′-OH ends (Fig. 4B upper panel). The in vitro data presented in Fig. 4B show that MenT1 could modify all five tRNAs in the presence of CTP, although some differences in intensity could be observed between these tRNA (Leu-3 and Ser-4 being less intensively modified), suggesting that some preference might exist. These data also further confirmed that CTP is the preferred substrate of MenT1 (Fig. 4C). Noticeably, in the presence of CTP or GTP, and to a lesser extent UTP, we found that the presence of the catalytic mutant MenT1 D41A substitution induces the formation of a shorter tRNA species (Fig. 4C), which we could identify by tRNA sequencing as Gly-3 cleaved of its last adenosine nucleotide (CCΔA, representing about 11% of the Gly-3 CCA ends) (Fig. S5B, Supplementary Datasheet 1), suggesting a 3′ to 5′ exonuclease activity of this non-toxic MenT1 derivative.

Finally, Gly-3 tRNA was used to investigate the importance of 3′ end nucleotides for recognition by MenT1. We found that deletion of the CCA with or without the discrimination base (ΔCCA and ΔUCCA), or the deletion or mutation of the last adenosine, abolished Gly-3 modification by MenT1 (Fig. 4D). In addition, mutations of the CC positions of the CCA end of the Gly-3 tRNA (i.e., CAA, CTA, CGA, ACA, TCA and GCA ends) exhibited little (CTA and GCA) or no detectable effect (CAA, CGA, ACA and TCA) on modification by MenT1 (Fig. 4E), which is in sharp contrast with the inhibition effect of mutation in the last adenosine (Fig. 4D). In order to investigate whether the tRNA acceptor stem alone was sufficient to be recognized and modified by MenT1, a Gly-3 tRNA acceptor stem construct and several of its variants with successive deletions of the UCCA-3′ end were tested for modification by MenT1 in the presence of [α-32P]- CTP (Fig. 4F). In this case, we found that the Gly-3 tRNA acceptor stem containing the wild-type CCA end could still be modified by MenT1. Consistent with the data obtained with the full-length Gly-3 tRNA, deletion of the last adenosine was sufficient to inhibit modification by MenT1 (Fig. 4F).

**MenA1 neutralizes MenT1 toxicity**

In contrast with MenA3 and MenA4, which belong to the COG5340 antitoxin family[27,41,42], MenA1 is a significantly shorter protein (68 amino acids) originally predicted to belong to the type I SymE RNase toxins[12]. We purified MenA1 and tested whether it could inhibit MenT1 activity in vitro, using both total RNA extracts and Gly-3 tRNA as substrates (Fig. 5A, B). Our results show that co-incubation of MenA1 with MenT1 prevents CMP transfer to tRNA, indicating that MenA1 can indeed inhibit MenT1 NTase activity in vitro.

MenA3 is an antitoxic kinase that is thought to inhibit MenT3 by phosphorylating residue S78 in the catalytic site of MenT3. In this case, alanine substitution S78A fully abolished MenT3 inhibition by MenA3[27]. However, MenA1 does not have any kinase domain detectable by sequence, and the corresponding T39A substitution in MenT1 was still inhibited by MenA1 (Fig. S6A), suggesting a different mode of antitoxicity. SEC analysis revealed that co-incubation of purified MenA1 and MenT1 led to the formation of a higher molecular weight complex of about 50.4 kDa (Fig. 5C), suggesting the existence of a

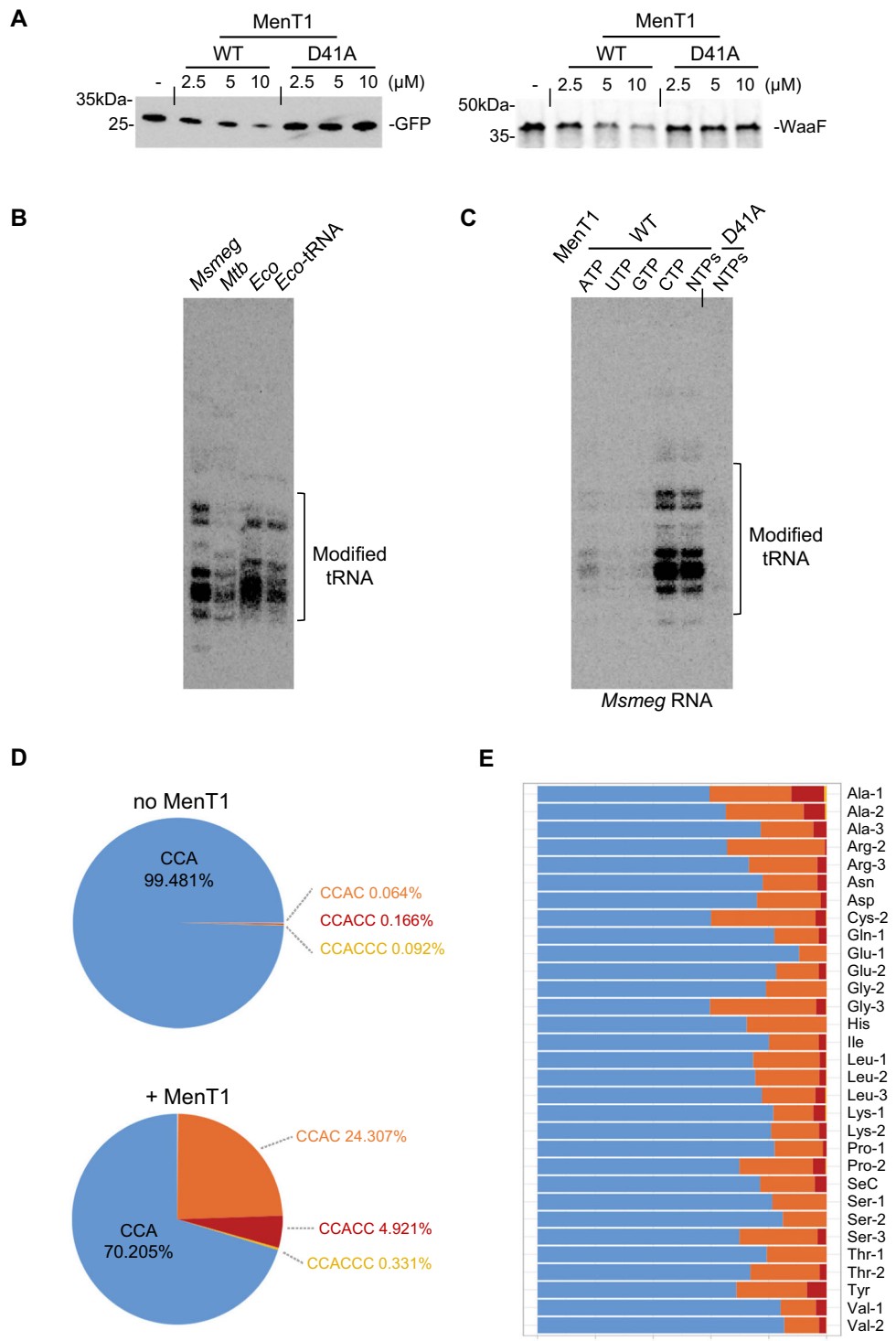

**Fig. 3 | MenT1 inhibits translation by modifying tRNAs in vitro. A** Total tRNA from *E. coli* was pre-incubated with MenT1 in vitro and subsequently used in a cell-free translation assay. Samples were separated on 4–20% SDS-PAGE gel and expression levels of both GFP and WaaF model substrates were determined using anti-GFP antibody and autoradiography, respectively. **B** tRNA modification by MenT1 was analysed by incubating 1 µg of total RNA from *M. smegmatis*, *M. tuber-culosis*, or *E. coli* or 100 ng of purified *E. coli* tRNA with MenT1 (5 µM) in the presence of [α–$^{32}$P] labelled NTPs at 37 °C for 2 h. Samples were separated on 6% Urea-PAGE gel and revealed by autoradiography. **C** Total RNA from *M. smegmatis* was incubated with MenT1 (5 µM) in the presence of [α–$^{32}$P] labelled ATP, UTP, GTP, or CTP

at 37 °C for 2 h. Representative results of triplicate experiments are shown. Source data are provided as a Source data file. **D** tRNA 3′-ends modified by MenT1 were mapped by incubating total RNA from *M. smegmatis* with or without MenT1 in the presence of CTP at 37 °C for 1 h; the final RNA and protein concentration was 0.5 µg.µl$^{-1}$ and 1 µg.µl$^{-1}$, respectively. Samples were then treated with a demethylase and the library was constructed as described in the methods section. The DNA was subsequently sequenced and the total modified tRNA reads were counted in samples with or without MenT1. **E** Modification of each specific tRNA form (**D**) are shown. Results are from two independent experiments.

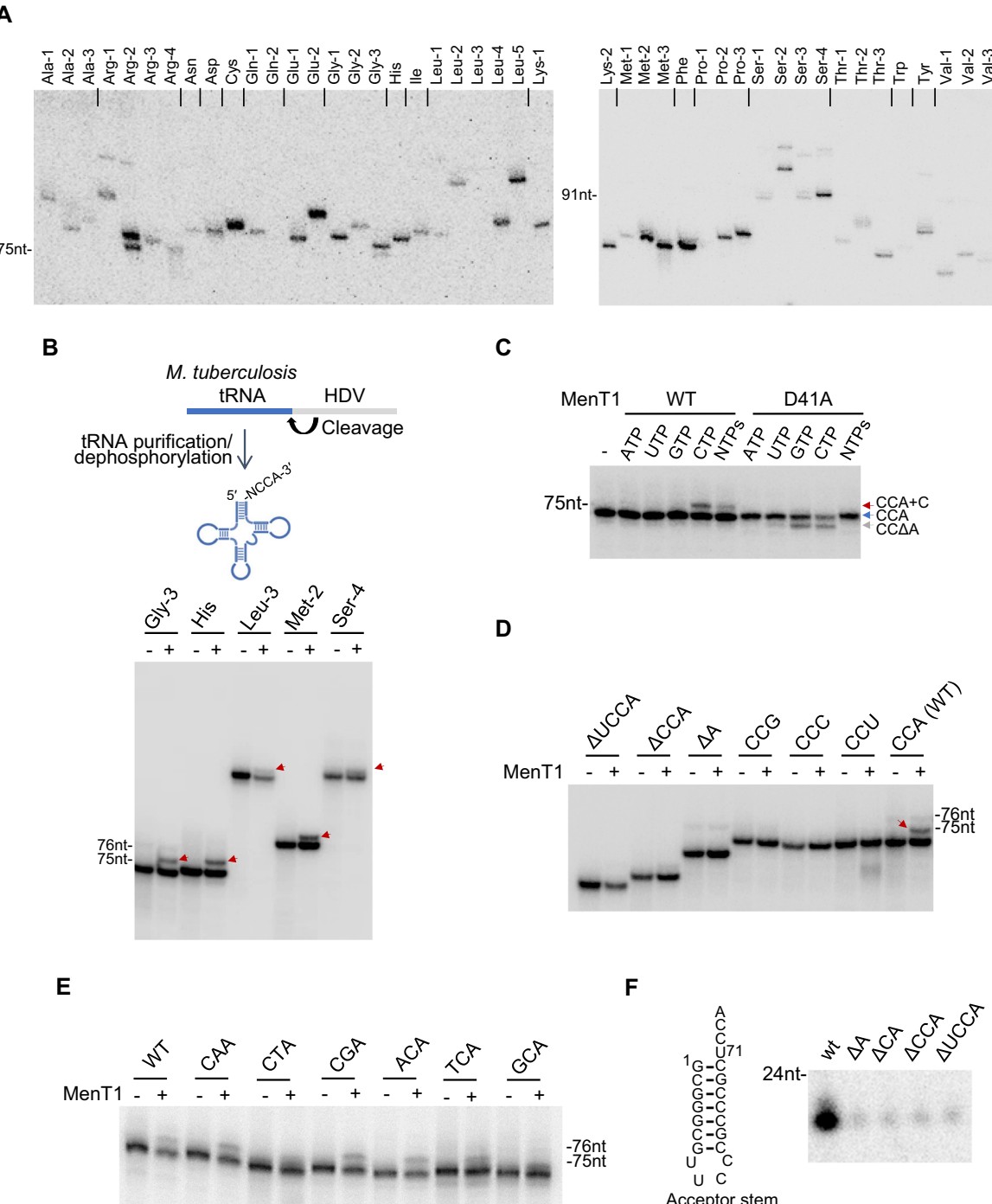

**Fig. 4 | MenT1 modifies in vitro transcribed tRNAs of *M. tuberculosis*. A** The 45 *M. tuberculosis* tRNAs were T7 transcribed in vitro and separately incubated with MenT1 in the presence of [α-³²P]-CTP at 37 °C for 2 h. tRNAs were separated on a 6% Urea-PAGE gel and revealed by autoradiography. This screening procedure of all 45 tRNA was performed once. **B** *M. tuberculosis* tRNAs with homogeneous 3′-ends were generated following ribozyme-mediated cleavage (upper scheme). Selected [α-³²P]-CTP labelled *M. tuberculosis* tRNAs were separately incubated with MenT1 (5 μM) for 4 h at 37 °C in the presence of unlabelled CTP. **C** [α-³²P]-CTP labelled *M. tuberculosis* tRNA Gly-3 was incubated with 5 μM of MenT1 WT or MenT1 D41A for 4 h at 37 °C in

the presence of unlabelled ATP, UTP, GTP, or CTP. The red arrows indicate the presence of cytidine extension. **D**, **E** MenT1 3′-end preference was determined by incubating [α-³²P]-CTP labelled *M. tuberculosis* tRNA Gly-3, 3′ truncated variants or 3′ single mutations with MenT1 (5 μM) for 4 h at 37 °C in the presence of unlabelled CTP. **F** The acceptor stem of Gly-3 tRNA modification by MenT1 in vitro. The synthesized acceptor stem of Gly-3 tRNA or 3′ trimming variants were incubating with MenT1 (5 μM) for 2 h at 37 °C in the presence of [α-³²P]-CTP. Representative results of triplicate experiments are shown. Source data are provided as a Source data file.

stable TA complex in which the MenT1 toxin is neutralized by its cognate antitoxin. We therefore co-incubated MenT1 and MenA1 overnight, re-purified the complex, and used the sample in crystallization trials. We subsequently determined the X-ray crystallographic structure of the MenAT1 complex, to a resolution of 1.44 Å (Fig. 5D, E,

Table 1). The MenAT1 crystal structure features two MenT1 protomers (MenT1α and MenT1β) bound to a MenA1 antitoxin monomer to form a MenT1α:MenA1:MenT1β complex (Fig. 5D, E). One and two gaps are present in the chains of MenT1α and MenT1β, respectively, occurring in external flexible loops. The MenA1 antitoxin protein consists of a

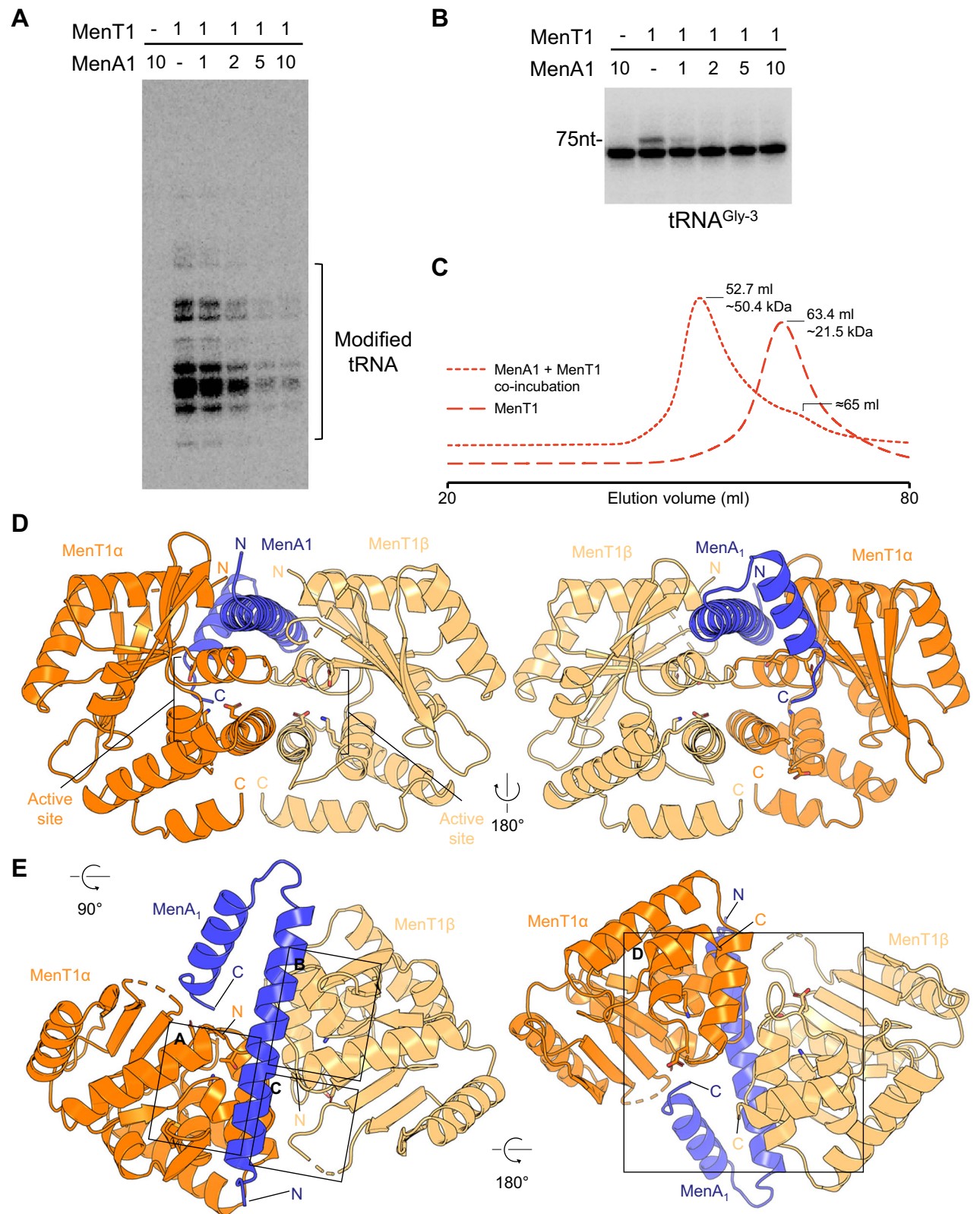

long central helix (α1) linked by a flexible loop to a shorter C-terminal helix (α2). The α2 helix folds back, partially towards the α1 helix, yet angled outwards, forming an asymmetrical, twisted "V" shape (Fig. 5D, E). Only 51 amino acids out of 68 were resolved in the MenA1 crystal structure. The 17 residues for which there was no electron density comprise the remainder of the MenA1 C-terminus, which is orientated towards the exterior of the crystal structure (Fig. 5D, E). Based on this

structure we engineered two truncated versions of MenA1, one containing the 51 amino acids resolved in the structure, and the other containing only the 32 amino acids of the large central helix (α1). Both truncated versions of MenA1 showed similar antitoxic activity to the full-length MenA1 when expressed in *M. smegmatis* (Fig. 6, Fig. S6). This suggests that, when overexpressed, the C-terminal helix and unresolved residues of MenA1 are not required for toxin inhibition.

**Fig. 5 | MenA1 and MenT1 form an asymmetric heterotrimeric complex. A** 1 µg total RNA of *M. smegmatis* was incubated with MenT1 (5 µM), along with [α-³²P]-labelled CTP, at 37 °C for 2 h, in the presence of increasing molar ratios of MenA1 antitoxin. Representative results of triplicate experiments are shown. **B** [α-³²P]-CTP labelled *M. tuberculosis* tRNA Gly-3 was incubated with 5 µM of MenT1 WT or MenT1 D41A at 37 °C for 4 h in the presence of unlabelled CTP and increasing molar ratios of MenA1 antitoxin. Representative results of triplicate experiments are shown. Source data are provided as a Source data file. **C** Purified MenT1 and MenA1 samples were combined and incubated overnight, then separated by Size Exclusion Chromatography (SEC) using a HiPrep 16/60 Sephacryl S-200 SEC column (Cytiva).

MenAT1 eluted at an elution volume of 52.7 ml, corresponding to a mass matching the predicted Mr of a MenT1:MenA1:MenT1 heterotrimeric complex. MenT1 eluted at 63.4 ml, corresponding to a mass matching monomeric MenT1. **D, E** Structure of the MenAT1 complex, with views from each side, above and below, shown as cartoons coloured orange (MenT1α), blue (MenA1), and light orange (MenT1β). N and C termini are indicated. To help compare their relative positions versus MenA1 the locations of active site residues identified in Fig. 2F (T39, D41, K137 and D152) are indicated, and the residues are shown as sticks, coloured red for oxygen, blue for nitrogen. Black boxes drawn in (**E**) correspond to regions shown in detail as panels Fig. 6A–D.

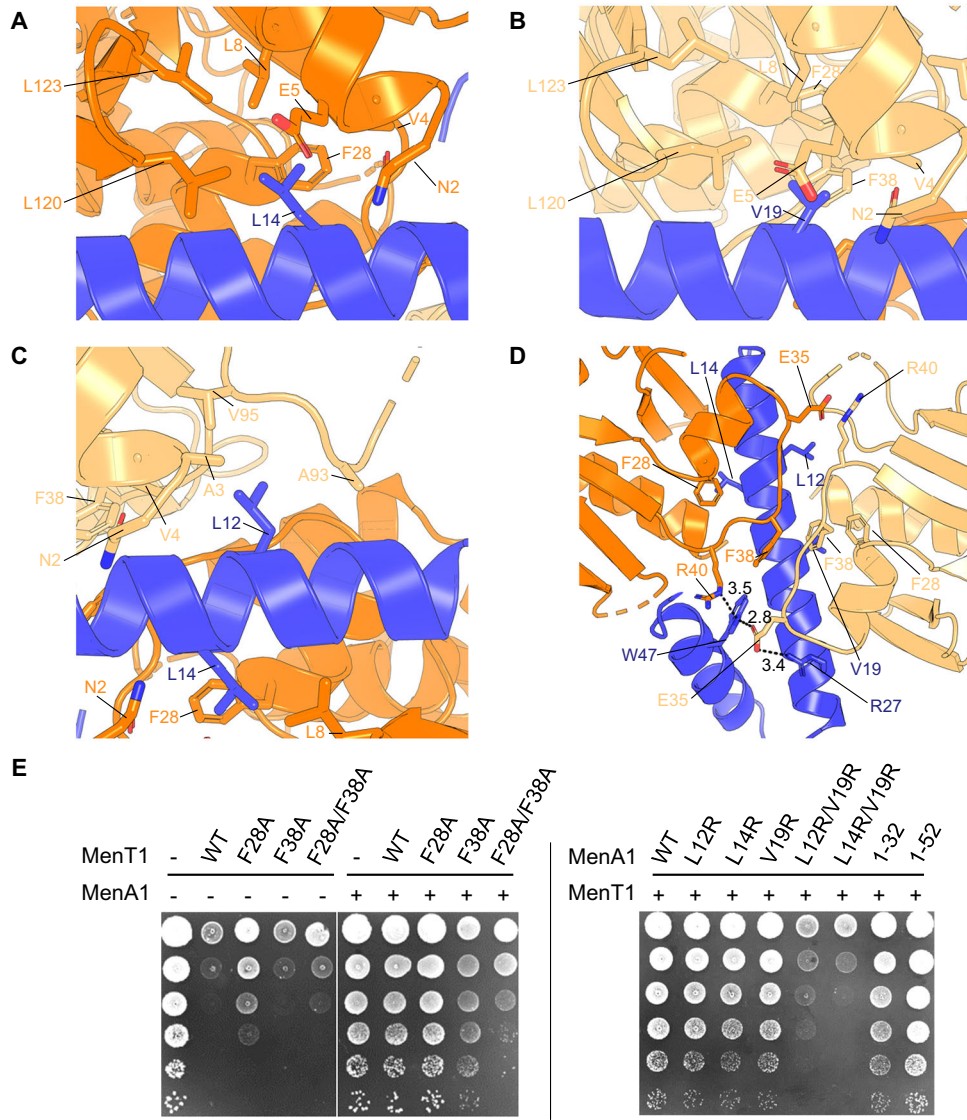

**Fig. 6 | Mutating MenA1-MenT1 interaction residues blocks antitoxicity.**
**A–C** Close-up views of boxed regions from (Fig. 5E), rotated to put MenA1 horizontal, coloured as per (Fig. 5E). **D** Close-up view of boxed region from (Fig. 5E), coloured as per (Fig. 5E). Black numbers indicate measured bond lengths in Å.
**E** Toxicity and antitoxicity assays were performed in *M. smegmatis* to study the effect of MenA1 on MenT1. Co-transformants of *M. smegmatis* containing (i) pGMC-

vector (-), MenT1 WT, or MenT1 F28A, F38A or F28A/F38A mutants, and pLAM-vector (-) or MenA1 WT (left panel), or (ii) MenT1 WT, and MenA1 WT, or MenA1 L12R, L14R, V19R, L12R/V19R, L14R/V19R, 1–32 or 1–52 mutants (right panel), were serially diluted and spotted on LB agar plates in the presence or absence of inducers (Atc, 100 ng.ml⁻¹ or Ace, 0.2%). The plates were incubated for 3 days at 37 °C. Representative results of triplicate experiments are shown.

The SEC trace from the combined purified MenT1 and MenA1 samples corroborates the formation of a heterotrimeric complex in solution (Fig. 5C), and MenA1 was confirmed for size by SDS-PAGE analysis (Fig. S7A). Within the complex structure, the resolved MenA1 antitoxin binds asymmetrically, running in opposing directions across

the same face of each MenT1 protomer (Figs. S7B, C). This brings the electropositive cavities of each MenT1 protomer into juxtaposition, suggesting that MenA1 prevents MenT1 toxicity by blocking the active site. Collectively, these data indicate that MenAT1 interactions differ from that of MenAT3 and MenAT4.

## MenA1 binds asymmetrically to form a heterotrimeric MenAT1 complex

The MenA1:MenT1 binding interfaces were examined to identify key residues for protein complex formation. Close-up views revealed three MenT1 hydrophobic pockets with which MenA1 interacts (Fig. 6A–C). Due to the asymmetric binding of MenA1, two identical MenT1 hydrophobic pockets face MenA1, one from each MenT1 protomer, binding opposite sides of MenA1 helix α1 (Fig. 6A, B). MenA1 residues L14 and V19 bind into pockets lined with L8, L120 and L123 from MenT1α and MenT1β, respectively (Fig. 6A, B). Meanwhile, residue L12 of MenA1 interacts with residues A3 and V4 of the MenT1β N-terminal α-helix, which form a hydrophobic region with MenT1β A93 and V95 from a linker loop connecting the β-strands (Fig. 6C). Relative to these three pockets aligning with what can be considered the sides of MenA1 helix α1, underneath MenA1 helix α1 there is a MenT1α:MenT1β interaction formed by loops that include F38, and the positions of these loops differs in each protomer (Fig. 6D). A close-up view of the interaction interface reveals that MenT1α F38 is pulled out by the asymmetry of MenA1 binding, allowing formation of a hydrophobic network comprising MenT1α F38, MenT1β F28 and MenT1β F38 stacked underneath MenA1 (Fig. 6D). This interaction network indicates the likely importance of these loops, and MenT1 residues F28 and F38, in MenA1 binding and subsequent MenAT1 complex formation. To verify such a hypothesis, we first constructed a set of alanine substitutions at these residues, i.e., F28A, F38A and F28A/F38A, and tested them in vivo by co-expressing MenT1 and MenA1 from compatible plasmids co-transformed in *M. smegmatis* (Fig. 6E). In this case, we found that MenA1 poorly inhibits MenT1 F28A/F38A and to a lesser extent MenT1 F38A when compared to MenT1 WT, suggesting that this region is indeed important for toxin neutralization. Note that the F28A single substitution also slightly affected MenT1 toxicity (Fig. 6E). In addition, we engineered several substitutions in the interacting surface of MenA1 and tested them in *M. smegmatis* (Fig. 6E). While the L12R, L14R or V19R single substitutions of MenA1 had no impact, the double L12R/V19R or L14R/V19R exhibited a strong effect on MenA1 activity (Fig. 6E), indicating that these regions are indeed important for MenT1 inhibition.

The MenAT1 crystal structure was submitted to PDBsum[36] for analysis. PDBsum calculated a buried surface area of 895 Å$^2$ between MenA1 and MenT1β, and identified a salt bridge formed between MenA1 R27 and MenT1β E35, supported by hydrogen bonding with MenA1 W47 (Fig. 6D). The rest of the calculated MenA1:MenT1β interface was formed through van der Waals interactions. For MenA1:MenT1α, PDBsum calculated a buried surface area of 1189 Å$^2$. The majority of the interface was again proposed to form through van der Waals interactions, with hydrogen bonding also formed through several interactions including again MenA1 W47 and MenT1α R40 (Fig. 6D). Note that as MenA1 W47 inserts between MenT1α R40 and MenT1β E35, a direct salt bridge forms between MenT1β R40 and MenT1α E35 at the reciprocal site on the other side of the complex (Fig. 6D). This range of distinct salt bridges and hydrogen bonding interactions between MenA1:MenT1α and MenA1:MenT1β highlight the asymmetrical binding of MenA1 to the MenT1 protomers. Collectively, the data support the formation of a heterotrimeric MenT1α:MenA1:MenT1β complex.

Aligning the structures of MenT1α and MenT1β from the MenAT1 structure produces an RMSD of 0.494 Å (1988 atoms), indicating a near-identical structural architecture. The only gross differences are the different positions of the F38 loop, and variations in the position of the C-terminal α-helix (Fig. S8A). Aligning the two MenT1 protomers from the MenT1 apo structure with both MenT1 protomers from the MenAT1 complex structure shows similarly little variation (Fig. S8B). If the MenT1 structure is superposed onto the MenAT1-structure, aligning through MenT1α, it is then clear that the second MenT1 protomers are found in very different relative positions

(Fig. S8C). Within this alignment, there is a clash between MenA1 and the second protomer of the MenT1 apo structure (Fig. S8C). This further indicates that within the MenT1 apo structure, the two protomers are not arranged in a way that is competent for interaction with MenA1, and so, as supported by SEC data (Fig. S3C), the presence of two protomers in the MenT1 apo crystals is merely a result of crystal packing. Collectively, these structures highlight conserved structural motifs within the MenAT family, amidst striking variations in how the toxins and antitoxins interact.

## tRNA modification by MenT-family toxins

Having characterised the in vitro tRNA NTase activity of MenT1 (Fig. 3), we then performed a similar analysis using the as-yet biochemically uncharacterized toxin, MenT4. When tested in the presence of labelled [α-$^{32}$P]-NTPs, in vitro incubation of MenT4 with total RNA extracts from *M. smegmatis* successfully showed the appearance of labelled products at the size of tRNAs (Fig. 7A). Unexpectedly, the tRNA NTase activity of MenT4 used GTP (Fig. 7A), and not CTP, as for MenT1 (Fig. 3).

Next, we used tRNA sequencing (Fig. S5) to examine which tRNAs had been modified by MenT4. Total RNA extracts of *M. smegmatis* (as used in Fig. 3B) were incubated with purified MenT4 in the presence of GTP and the 3′-OH RNA-seq libraries were generated and sequenced (Fig. 7B). In the absence of toxin, again the majority (over 99.8%) of the tRNA ends possessed the expected CCA acceptor sequence, whereas in the presence of toxin MenT4, ~2% had acquired extra guanosines (one or two). Though this shows clear in vitro activity for MenT4, it appears to be lower than for MenT1 (Fig. 3D). Finally, MenT4 was shown to modify 34 of the 37 tRNA identified in the sequencing experiment (Fig. 7B, Supplementary Datasheet 1), suggesting that MenT4 also exhibits no strong preference for specific tRNA. Collectively, these data show MenT toxins have nucleotide preferences, can vary in the number of bases added to target tRNAs in vitro, and that tRNA NTase activity can be observed across the MenT family.

## Discussion

This work has characterized the MenAT1 (SymE-like/NTase) TA pair of the major human pathogen *M. tuberculosis*, establishing the mechanism of toxicity of MenT1 and its interaction with the MenA1 antitoxin. We show that MenT1 is a bona fide NTase that prevents aminoacylation of tRNAs by specifically adding cytidines to their 3′ CCA ends, leading to *M. tuberculosis* growth inhibition. Furthermore, we have solved the crystal structures of both the toxin alone and in complex with its antitoxin, revealing a novel interaction for MenAT TA systems. Finally, we show that tRNA modification is a hallmark of MenT toxins.

MenT1 is toxic when expressed in *M. smegmatis* or *M. tuberculosis*, but has no detectable effect on *E. coli* growth, even when over-expressed to high levels[25], and does not inhibit translation in *E. coli* cell-free assays unless pre-incubated with significantly lower concentrations of total tRNAs (Fig. 3A). Although the reason for MenT1 toxicity only manifesting in vivo within mycobacteria is unknown, differences in the abundance of some, if not all, tRNAs might be critical. While there is no comparative study about tRNA abundance in these bacteria, the *E. coli* genome possesses multiple tRNA genes, 88 in total, and the *M. tuberculosis* genome encodes only 45, which could explain the observed difference in activity[38,43]. In addition, differences in tRNA modification, aminoacyl-tRNA synthetases, repair mechanisms (e.g., RNase PH) and maturation (e.g., CCA-adding enzyme), or the lack of specific partners in *E. coli*, could also play a role. For example, it is not known whether native tRNA modifications that occur during their maturation could influence their recognition and modification by MenT1. The fact that our model Gly-3 tRNA was more heavily modified by MenT1 when purified from in vivo cell extracts (~40%) than from in vitro synthesis (~13%) suggests that mature tRNA could be more efficiently targeted by MenT1.

In contrast with MenT1, the MenT3 toxin of *M. tuberculosis* is toxic in *E. coli* and efficiently inhibits translation in *E. coli* cell-free assays without the need of pre-incubation or dilution of the tRNA pool[25]. Remarkably, while MenT1 can target most, if not all, tRNAs in vitro, MenT3 showed significant preference for serine tRNA isoacceptors[25], suggesting that targeting specific sets of tRNA in vivo might be more effective than randomly targeting all tRNA. This is in agreement with the fact that MenT3 is also more toxic than MenT1 both in *M. smegmatis* and in *M. tuberculosis,* i.e., (i) *menT3* deletion could only be obtained in the presence of a *trans* copy of *menA3* and (ii) its cloning was possible only when a weaker Shine-Dalgarno sequence was used[25]. Intriguingly, serine tRNAs were recently shown to be the less frequently charged tRNAs of *M. tuberculosis* when at an in vivo steady state, which would make them even more accessible to MenT3 and thus further exacerbate its deleterious effect[44]. Whether specific sequences or structural elements of serine tRNA would contribute to such preference by MenT3 remains unknown. Though structural comparisons identified a possible tRNA binding mode for MenT toxins (Fig. S4), this gives no detailed structural rationale for why MenT3 but not MenT1 or MenT4, shows such a preference in vitro. Finally, MenT4 activity also showed no apparent preference for specific tRNA and a less robust toxicity than MenT3[25], thus highlighting a continuum of target selection throughout the MenT toxin family. One attractive possibility is that toxins with low specificity towards the host tRNA, i.e., MenT1 and MenT4, would specifically target and prevent aminoacylation of phage-encoded tRNA[8,45], thus acting as phage defence systems.

Targeting of specific tRNAs has been observed for other toxin families. While acetyltransferase toxins were shown to preferentially acetylate the primary amine group of charged (i) tRNA^Gly in the case of TacT[46–48] and (ii) tRNA^Ile, tRNA^Val and tRNA^Met for ItaT[49,50], the HEPN toxin was shown to cleave the last four nucleotides of the 3' end of only a subset of tRNAs[51]. The ToxSAS toxins PhRel, CapRel and FaRel2 transfer a pyrophosphate moiety from ATP to the tRNA CCA end[52], the CreT RNA toxin sequesters tRNA^Arg[53] and the canonical serine-threonine kinase HipA phosphorylates glutamyl-tRNA synthetase causing inhibition of glutamyl-tRNA charging[54]. In *M. tuberculosis* only TacT (Rv0919), which targets glycyl-tRNAs, and VapC4, which cleaves a single site within the anticodon sequence of tRNA^Cys [16], were identified so far. Together, these data suggest that in many cases, targeting specific sets of tRNAs might be a hallmark for TA systems.

Our work shows that the integrity of a tRNA CCA 3'-end is crucial for MenT1 activity in vitro, indicating that MenT1 could likely target mature uncharged tRNAs in vivo, either from newly synthesized tRNA or deacyl-tRNA released during elongation. How *M. tuberculosis* would respond to such a reduced pool of translation-competent tRNA is not known. Interestingly, it was previously shown that in the case of the GCN5-related N-acetyltransferase (GNAT) toxins of *Salmonella*[48] and *M. tuberculosis*[47], the noxious amino acid acetylation of glycyl-tRNAs that leads to translation inhibition could be rescued by the peptidyl-tRNA hydrolase Pth, which efficiently cleaves the acetylated amino acid from tRNA, thus unblocking protein synthesis. Maturation of tRNA CCA 3'-ends occurs through exo- and/or endo-nucleolytic pathways involving several ribonucleases, including RNase Z, RNase PH and/or RNase T[55,56], and CCA-adding enzymes[56,57]. Although we have no evidence that cytidine-modified tRNA can be recognized and repaired by these pathways in *M. tuberculosis*, it is reasonable to assume that the success of NTase toxins could, at least in part, rely on the repairing capacity of the host. Two-thirds (30/45) of the tRNA genes of *M. tuberculosis* do not contain a CCA motif in their sequences (while all *E. coli* tRNA genes do have CCA) and need to be processed by CCA-adding enzymes. This suggests that in the presence of activated MenT1, maturation enzymes, especially CCA-adding enzymes, could be overwhelmed by the new pool of accumulating deacylated CCA +

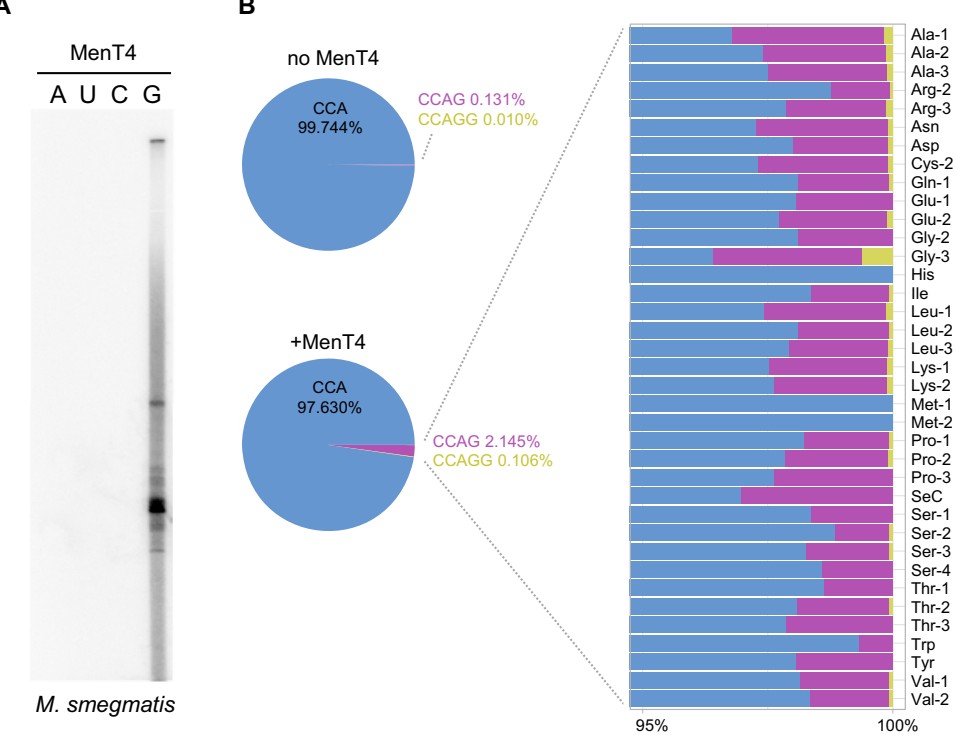

**Fig. 7 | MenT4 has preference for GTP and modifies tRNAs. A** 1 μg of total RNA from *M. smegmatis* was incubated with MenT4 (5 μM) in the presence of [α–³²P] labelled ATP, UTP, GTP, or CTP at 37 °C for 3 h. Total RNA were separated on a 6% Urea-PAGE gel and revealed by autoradiography. Representative results of duplicate experiments are shown. **B** To study the activity of MenT4 on tRNA substrates, tRNA 3' mapping was performed using 5 μg of total RNA from *M. smegmatis* incubated with 10 μg of MenT4 in the presence of GTP at 37 °C for 2 h. A tRNA sequencing library was then generated and sequenced. The percentage of modification for total tRNA (left) and for each tRNA identified (right) is shown. Representative results of duplicate experiments are shown.

C(n) tRNA 3′-ends. More work is warranted to shed light on possible rescue mechanisms.

Within the MenAT1 complex, identical sites on each of two MenT1 protomers are bound by independent sites on the single MenA1 protomer, due to asymmetric binding across the faces of each toxin. There is precedent for heterotrimeric toxin-antitoxin complexes. Examples can be drawn from the Kid family of toxins[58], including PemIK[59] and MazEF[60]. In PemIK from *Staphylococcus aureus*, a PemK toxin dimer is blocked by a loose strand (but not a structured α-helix) from a single PemI antitoxin. Similarly, though the heterohexameric structure of MazEF[60] from *Bacillus subtilis* shows symmetrical TA complex architecture, the individual binding of one MazE antitoxin to a dimer of MazF toxins shows a heterotrimer blocked by an antitoxin α-helix. Though loosely analogous to the MenAT1 complex, there are significant differences in these examples, such as the two MenT1 toxins not themselves interacting as a dimer, and instead existing as monomers in solution that are brought together in complex by MenA1. In contrast, the PemK and MazF toxins exist and operate as dimeric RNases. Furthermore, the MenT and Kid family folds are unrelated. As such, MenAT1 appears to form a novel complex that is suggested to both sequester and block MenT1 toxins using asymmetric antitoxin binding. This activity of MenA1 also demonstrates that multiple routes for inhibition can be found within this single family of MenAT TA systems.

This study has revealed heterogeneity in multiple aspects of MenAT family activities, such as target tRNA preferences, use of nucleotides as substrates and modes of antitoxin inhibition. How these reflect in vivo roles for each MenAT system, and the impact of *menAT* deletion mutants in *M. tuberculosis*, remains to be discovered.

## Methods

### MenAT1 sequence analysis

Analysis of gene neighbourhoods for *rv0078B* (*menA1*) was performed using default settings in FlaGs (http://www.webflags.se/). Output sequences for MenA1 and cognate MenT1 homologues were then used to perform sequence alignments using MUSCLE (https://www.ebi.ac.uk/Tools/msa/muscle/), then formatted in Jalview (https://www.jalview.org/), sorting by pairwise alignment. Residues of interest were then highlighted using ESPript (https://espript.ibcp.fr/ESPript/ESPript/).

### Bacterial strains

*E. coli* strains BL21(λDE3) and BL21 (λDE3) AI (Novagen), DH5α (Invitrogen), BL21 (λDE3) Δ*slyD*[25] and DLT1900[61], and *M. smegmatis* mc²155 (ATCC 700084) have been described. *E. coli* were routinely grown at 37 °C in LB, M9 or 2xYT medium supplemented, when necessary, with kanamycin (Km, 50 µg.ml⁻¹), chloramphenicol (Cm, 34 µg.ml⁻¹), ampicillin (Ap, 50 µg.ml⁻¹), spectinomycin (Sp, 50 µg.ml⁻¹), isopropyl-β-D-thiogalactopyranoside (IPTG, 1 mM), L-arabinose (L-ara, 0.1% w/v) or D-glucose (glu, 0.2% w/v). *M. smegmatis* mc²155 strains were routinely grown at 37 °C either in LB or 7H9 medium (Difco) supplemented, when necessary, with Km (10 µg.ml⁻¹) or streptomycin (Sm, 25 µg.ml⁻¹). *M. tuberculosis* wild-type strain H37Rv (ATCC 27294) and mutant strains were routinely grown at 37 °C in 7H9 medium, (Middlebrook 7H9 medium, Difco) supplemented with 10% albumin-dextrose-catalase (ADC, Difco) and 0.05% Tween 80 (Sigma-Aldrich), or on complete 7H11 solid medium (Middlebrook 7H11 agar, Difco) supplemented with 10% oleic acid-albumin-dextrose-catalase (OADC, Difco). When required, mycobacterial growth media were supplemented with, hygromycin (Hy, 50 µg.ml⁻¹), Sm (25 µg.ml⁻¹), zeocin (Zeo, 25 µg.ml⁻¹), acetamide (Ace, 0.2% w/v) or anhydrotetracycline (Atc, 100 or 200 ng.ml⁻¹).

### Construction of *M. tuberculosis* mutant

Mutant strains of *M. tuberculosis* H37Rv was constructed by allelic exchange using recombineering[62], as described[63]. Two DNA fragments (-0.5-kb) flanking the genes of interest were amplified by PCR with PrimeSTAR GXL DNA polymerase (Takara bio), *M. tuberculosis* H37Rv genomic DNA as template and appropriate primer pairs (Supplementary Table S1). A three-fragment PCR fused these two fragments to a Zeo resistance cassette and the resulting allelic exchange substrate (AES) was recovered by agarose gel purifications and used to transform H37Rv carrying plasmid pJV53H, a hygromycin-resistant pJV53-derived plasmid expressing recombineering enzymes[62]. H37Rv/pJV53H was grown in complete 7H9 medium supplemented with Hy until mid-log phase, treated by 0.2% Ace (Sigma-Aldrich) overnight at 37 °C and electrotransformed with 100 ng of the linear AES. After 48 h incubation at 37 °C in complete 7H9 without antibiotic, the transformation mix was plated on agar plates supplemented with Zeo and incubated 21 days at 37 °C. Zeo resistant clones were streaked on the same medium, and single colonies were grown in complete 7H9 without antibiotic and verified to carry the expected allele replacement by PCR on chromosomal DNA, using external primers. Spontaneous loss of the pJV53H plasmid was obtained by serial rounds of culture without antibiotics and phenotypic tests for resistance to Zeo and sensitivity to Hy.

### Plasmid constructs

Plasmids pMPMK6[64], p29SEN[65], pGMCS[66], pLAM12[62], pETDuet-1 and pET15b (Novagen) have been described. All the primers used to construct the plasmids are described in Supplementary Table S1. Plasmid pK6-MenT1 was constructed as follows. *Rv0078A* was PCR amplified from the *M. tuberculosis* H37*Rv* genome using primers Rv0078A EcoRI-For and Rv0078A HindIII-Rev, and cloned as EcoRI/HindIII fragments into EcoRI/HindIII digested pMPMK6. For p29-MenA1, *rv0078B* was PCR amplified from *M. tuberculosis* H37*Rv* genome using primers Rv0078B EcoRI-For and Rv0078B HindIII-Rev, and cloned as EcoRI/HindIII fragments into EcoRI/HindIII digested p29SEN.

Plasmid pGMC-MenT1 and pGMC-MenT1$_{His}$ were obtained following PCR amplification of *rv0078A* with primers Rv0078A In-Fusion-For and Rv0078A In-Fusion-Rev, and *rv0078A*$_{His}$ with primers Rv0078AHis In-Fusion-For and Rv0078A In-Fusion-Rev using pK6-MenT1 as template, and cloned by homologous recombination in linearized pGMCS plasmid using In-Fusion HD Cloning Kits (Takara Bio). Plasmid pGMC-MenA1T1, containing the *rv0078B-rv0078A* operon, was also obtained by In-Fusion using primers Rv0078B In-Fusion-For and Rv0078A In-Fusion-Rev and *M. tuberculosis* H37*Rv* genome as template. Plasmids pGMC-MenT1 with T39A, D41A, K137A, D152A, F28A, F38A or F28A/F38A substitutions were constructed by QuikChange site-directed mutagenesis using appropriate primers and pGMC-MenT1 as template.

Plasmid p29-MenA1 was used as template to construct pLAM-MenA1 and pLAM-MenA1$_{His}$. In this case, NdeI/EcoRI digested fragments of *rv0078B* and *rv0078B*$_{His}$ were cloned into pLAM12 vector digested with the same enzymes. Plasmid pLAM-MenA1 was then used as template to construct the pLAM-MenA1 L12R, L14R, V19R, L12/V19R, or L14/V19R substitutions by QuikChange site-directed mutagenesis using appropriate primers. Truncated MenA1 N1-32 and N1-52 were amplified from pLAM-MenA1 using primers Rv0078B NdeI-For and Rv0078B N1-32 EcoRI-Rev, or Rv0078B NdeI-For and Rv0078B N1-52 EcoRI-Rev, respectively, and cloned into pLAM vector digested with NdeI/EcoRI.

The pET15b vector derivatives used in this work were constructed as follows. To construct pET15b-MenT1$_{His}$, *rv0078A* was digested from pK6-MenT1 and cloned as an NdeI/BamHI fragment into pET15b digested with the same enzymes. Plasmid pET15b-MenT1$_{His}$ was used as template to construct pET15b-MenT1$_{His}$ D41A substitution by QuickChange. For pET-MenA1$_{His}$ (MenA1 with an N-terminal His$_6$-Ser-Ser-Gly-tag), *rv0078B*$_{His}$ was PCR amplified from p29-MenA1 template using primers Rv0078B$_{His}$ NcoI-For and Rv0078B$_{His}$ BamHI-Rev, and cloned

as an NcoI/BamHI fragment into pETDuet-1 digested with the same enzymes.

The pUC57-T7-His-HDV, pUC57-T7-Leu-3-HDV, pUC57-T7-Met-2-HDV and pUC57-T7-Ser-4-HDV plasmids containing tRNA-HDV fusion under the control of a T7 promoter were synthesized by Genewiz (Azenta Life Sciences). The sequences of the T7-His-HDV, T7-Leu-3-HDV, T7-Met-2-HDV and T7-Ser-4-HDV fragments are given in Supplementary Table S1. To prepare DNA template for transcription with T7 polymerase, T7-tRNA-HDV fragments were PCR amplified using primers T7-For and HDV-Rev for T7-His-HDV and T7-Met-2-HDV or HDV short-Rev for T7-Leu-3-HDV and T7-Ser-4-HDV. T7-Gly-3-HDV, T7-Gly-3ΔA-HDV, T7-Gly-3ΔCA-HDV, T7-Gly-3ΔCCA-HDV and T7-Gly-3ΔTCCA-HDV was constructed by In-fusion PCR and the primers were listed in Supplementary Table S1. To obtain T7-Gly-3CAA-HDV, T7-Gly-3CTA-HDV, T7-Gly-3CGA-HDV, T7-Gly-3ACA-HDV, T7-Gly-3TCA-HDV and T7-Gly-3GCA-HDV, the corresponding plasmid pGMC-T7-Gly-3-HDV mutations were constructed by Quickchange using pGMC-T7-Gly-3-HDV as template which was obtained following PCR amplification of T7-Gly-3-HDV with primers T7-Gly-3-HDV In-Fusion-For and T7-Gly-3-HDV In-Fusion-Rev using T7-Gly-3-HDV as template, and cloned by homologous recombination in linearized pGMCS plasmid using In-Fusion HD Cloning Kits (Takara Bio).

To construct plasmids pTRB617 (MenA1) and pTRB629 (MenT1) for protein expression, the Ligation Independent Cloning (LIC) site was first amplified from plasmid pSAT1-LIC[25], using primers TRB1460/TRB1462 (Supplementary Table S1) and inserted into pBAD30[67], resulting in plasmid pTRB550. The *rv0078B* gene was then amplified by PCR from *M. tuberculosis* H37Rv genomic DNA using primers TRB1699/TRB1700, and *rv0078A* was amplified from pET-MenT1$_{His}$ using primers TRB1701/TRB1702, and both coding sequences were cloned by LIC into pTRB550 to create pTRB617 and pTRB629, respectively. The pTRB550 expression vector adds a cleavable N-terminal His$_6$-SUMO tag.

## Bacterial growth assays

In vivo toxicity assays in *E. coli* were performed as follows. *E. coli* DLT1900 was grown in LB medium at 37 °C, co-transformed with pMPMK6-vector or pK6-MenT1, and p29SEN-vector or p29-MenA1 and grown overnight at 37 °C on LB Km, Ap, glu agar plates. Transformants were gown to mid-log phase in LB Km Ap medium, serial diluted and spotted on LB Km Ap agar plates with or without L-ara and/or IPTG. Plated were incubated at 37 °C. In vivo toxicity assays in *M. smegmatis* were performed as follows. Cultures of mc²155 strain grown in LB medium at 37 °C were co-transformed with the integrative pGMC-vector, pGMC-MenT1 and with pLAM12-vector, pLAM-MenA1 and selected on LB Km Sm agar plates for 3 days at 37 °C in the presence or absence of Atc (100 ng.ml⁻¹) and Ace (0.2%) for toxin and antitoxin expression, respectively. *M. tuberculosis* toxicity assays were performed as follows. *M. tuberculosis* strains H37Rv or H37RvΔ(*menA1-menT1*)::Zeo$^R$ were transformed by electroporation with ~50 ng of plasmids pGMCS vector, pGMC-MenA1-MenT1, pGMC-MenT1, or pGMC-MenT1 variants with T39A, D41A, K137A or D152A substitutions. After 3 days of phenotypic expression in 7H9 ADC Tween at 37 °C, the transformation mix was divided into two halves. One half was plated on 7H11 OADC with Sm and the other half was plated on the same medium supplemented with Atc (200 ng.ml⁻¹). Plates were imaged after 20 days of incubation at 37 °C.

## Protein purification

To purify MenT1 and its derivatives, strain BL21(λDE3) AI transformed with pET15b-MenT1$_{His}$ was grown to an OD$_{600}$ of ~0.4 at 37 °C, 0.2% L-ara was added and the culture immediately incubated overnight at 22 °C. Cultures were centrifuged at 5000 × *g* for 10 min at 4 °C, pellets were resuspended in Lysis Buffer (20 ml for 1 litre of cell culture) and incubated 30 min on ice. Lysis was performed using the One-shot cell

disrupter at 1.5 KBar (One shot model, Constant Systems Ltd). Lysates were centrifuged for 30 min at 30,000 × *g* in 4 °C and the resulting supernatants were gently mixed with Ni-NTA Agarose beads (Qiagen) pre-equilibrated with lysis buffer (25 mM Na-P, 200 mM NaCl, 20 mM imidazole) at 4 °C for 30 min in a 10 ml poly-prep column (Bio-Rad). The column was then stabilized for 10 min at 4 °C, washed three times with 10 ml of lysis buffer, and proteins were eluted with elution buffer (25 mM Na-P, 200 mM NaCl, 250 mM imidazole). 500 µl elutions were collected and PD MiniTrap G-25 columns (GE Healthcare) were used to exchange buffer (25 mM Na-P, 200 mM NaCl, 10% glycerol) and proteins were concentrated using vivaspin 6 columns with a 5000 Da cut off (Sartorius). To remove the His-tag, thrombin was incubated with the protein at 4 °C overnight. Following NTA and streptavidin addition, the cleaved His-tag and thrombin were washed out. Proteins were stored at −80 °C until further use.

For expression of MenA1 and MenT1 for crystallization, *E. coli* BL21 (λDE3) Δ*slyD* was transformed with pTRB617 and pTRB629, respectively. Single colonies were used to inoculate 25 ml of LB supplemented with Ap, for overnight growth at 37 °C with 200 rpm shaking. For large scale expressions, overnight starter cultures were re-seeded at a 1:100 v/v ratio into 12 × 2 L baffled flasks containing 1 L 2xYT and Ap. Cells were initially grown at 37 °C with 175 rpm shaking until an OD$_{600}$ of 0.3 was achieved, then the temperature was reduced to -25 °C until the expression culture reached an OD$_{600}$ of 0.55. At this point, the temperature was turned down to 16 °C, L-ara (0.1% w/v) was added to induce protein expression, and the culture was grown overnight with shaking at 175 rpm. Cells were then pelleted at 4200 × *g* for 30 min at 4 °C. Cell pellets were resuspended in 50 ml of ice-cold A500 (20 mM Tris–HCl pH 7.9, 500 mM NaCl, 10 mM imidazole and 10% glycerol), lysed via sonication, then clarified by centrifugation at 45,000 × *g* at 4 °C for 30 min. All clarified cell lysates were passed over a 5 ml HisTrap HP column (Cytiva), and washed with 50 ml of A500, followed by ten column volumes of A100 (20 mM Tris–HCl pH 7.9, 100 mM NaCl, 10 mM imidazole and 10% glycerol), then eluted directly onto a HiTrap Q HP anion exchange chromatography (AEC) column (Cytiva) with B100 (20 mM Tris–HCl pH 7.9, 100 mM NaCl, 250 mM imidazole and 10% glycerol). The Q HP column was transferred to an Äkta Pure (Cytiva), washed with 3 column volumes of A100, then proteins were eluted using a gradient from 100% A100 to 100% C1000 (20 mM Tris–HCl pH 7.9, 1000 mM NaCl, 10 mM imidazole and 10% glycerol). Äkta fractions containing the target protein, as indicated by the AEC chromatogram peak, were first analysed and verified by SDS-PAGE. These were then pooled and incubated overnight at 4 °C with human sentrin/SUMO-specific protease 2 (hSENP2) to cleave the His$_6$-SUMO tag from the target protein. The next day, the sample was passed through a second HisTrap HP column and the flow-through containing untagged target protein was collected. Prior to purification by size exclusion chromatography (SEC), the MenA1 antitoxin and MenT1 toxin samples were each divided into two. One sample each of MenA1 and MenT1 were directly mixed and co-incubated overnight at −4 °C. The MenT$_1$ sample and the mixed MenA$_1$-MenT$_1$ samples were then concentrated and run over a HiPrep 16/60 Sephacryl S-200 SEC column (Cytiva) connected to the Äkta Pure, using Sizing buffer (20 mM Tris–HCl pH 7.9, 500 mM KCl and 10% glycerol). Äkta fractions corresponding to the SEC chromatogram peak were analysed by SDS-PAGE, confirmed to contain target protein(s), then pooled and concentrated. Purified protein was either flash-frozen in liquid N$_2$ for storage at −80 °C or dialysed overnight at 4 °C into Crystal buffer (20 mM Tris–HCl pH 7.9, 200 mM NaCl and 2.5 mM DTT) for crystallographic studies. Crystallization samples were quantified using a NanoDrop 2000 Spectrophotometer (Thermo Fisher Scientific) and stored on ice, then either used immediately or flash-frozen in liquid N$_2$ for storage at −80 °C. Frozen crystallization samples still formed usable crystals at least 15 months after storage. Purified MenT$_4$ was obtained as previously described[25].

## Protein crystallization and structure determination

MenA1:MenT1 and MenT1 protein samples were concentrated to 12 mg.ml$^{-1}$ in Crystal buffer (see above) and crystallization screens were performed using a Mosquito Xtal3 robot (SPT Labtech), setting 200:100 nl and 100:100 nl protein:condition sitting drops. After initial screening, MenA1:MenT1 formed a single large cuboid crystal in condition F1 (0.2 M sodium fluoride, 0.1 M Bis-Tris propane pH 6.5 and 20% w/v PEG 3350) of PACT Premier Eco HT-96 (Molecular Dimensions), and MenT1 formed thin square crystals in conditions E8 (1.8 M lithium sulfate and 0.1 M Tris pH 7.5) and G8 (1.8 M lithium sulfate and 0.1 M Tris pH 8.5) of Clear Strategy II Eco HT-96 (Molecular Dimensions). To harvest, 20 μl of condition reservoir was added to 20 μl of Cryo buffer (25 mM Tris–HCl pH 7.9, 187.5 mM NaCl, 3.125 mM DTT, 80% glycerol) and mixed quickly by vortexing; this mixture was then added to the protein crystal drop at a 1:1 v/v ratio. After the addition of cryoprotectant, crystals were immediately extracted using a nylon loop and transferred to a uni-puck stored in liquid N$_2$.

Diffraction data were collected at Diamond Light Source on beamlines I24 (MenT1) and I04 (MenA1:MenT1), using Diamond Light Source's "Generic Data Acquisition" (opengda.org) client, and iSpyB (R2.3) to visualize the data. A single 720° dataset was collected for MenT1 at 0.9795 Å. Two 360° datasets were collected for MenA1:MenT1 at 0.9795 Å and merged using iSpyB (Diamond Light Source). Diffraction data were processed with XDS[68], and then AIMLESS from CCP4[69] was used to corroborate the spacegroups. The crystal structure of MenA1:MenT1 was solved ab initio using ARCIMBOLDO[70]. The MenT1 structure was solved by PHASER MR[71], with MenT1α from the MenA1:MenT1 structure used as a search model. All solved crystal structures were further built using BUCCANEER[72] and REFMAC in CCP4[69], then iteratively refined and built using PHENIX[73] and COOT[74], respectively. The quality of the final model was assessed using COOT and the wwPDB validation server[75]. Structural figures, including alignments and superpositions, were generated using PyMol[76].

## Cell-free protein synthesis

A cell-free transcription/translation coupled assay (PURE system, Protein synthesis Using Recombinant Elements, NEB) was used to monitor the effect of the MenT1 toxin and its derivatives on protein synthesis, as described for MenT3[25]. Template DNA of *gfp*[77] and *waaF* (P37692) were separately added according to the manufacturer's instructions to the PURE system in the presence or absence of the toxin. Protein synthesis was performed for 2 h at 37 °C, samples were separated on SDS-PAGE (4–20% miniprotein TGX gels from Bio-Rad) and analysed by autoradiography (WaaF) or by western blots (GFP). In the case of GFP, blots were probed with monoclonal anti-GFP antibody (ThermoFisher MA5-15256; dilution 1/3000), detected using HRP-conjugate anti-mouse IgG (H + L) secondary antibody (Promega W4021; dilution 1/2500) and visualized by Image Lab software (Bio-Rad). To preincubate MenT1 with tRNA, a PURExpress Δ(aa, tRNA) kit (NEB, E6840S) was used. 0.33 μl tRNA from the kit was incubated with 0, 2.5, 5, 10 μM MenT1 for 3 h at 37 °C, then the assay was performed as per the manual, except 1.5 μl pre-incubated tRNA was used in a 5 μl reaction volume.

## In vitro transcription of tRNAs

tRNAs were obtained following in vitro transcription of PCR templates containing an integrated T7 RNA polymerase promoter sequence. Primers for *M. tuberculosis* tRNAs are given in Supplementary Table S1. The T7 RNA polymerase in vitro transcription reactions were performed in 25 μl total volume with a 5 μl nucleotide mix of 2.5 mM NTPs (Promega). 50 to 100 ng of template were used per reaction with 1.5 μl rRNasin 40 U.ml$^{-1}$ (Promega), 5 μl 5x optimized transcription buffer (Promega), 2 μl T7 RNA polymerase (20 U.ml$^{-1}$) and 2.5 μl 100 mM DTT at 37 °C for 2 h[25]. The tRNA products were isolated using trizol[78] and

stored at a final concentration of 100 to 200 ng.μl$^{-1}$, as monitored by NanoDrop.

## Nucleotide transfer assay

MenT1 NTase activity was assayed in 10 μl reaction volumes containing 20 mM Tris–HCl pH 8.0, 10 mM MgCl$_2$ and 1 μCi.μl$^{-1}$ of radiolabelled rNTPs [α–$^{32}$P] (Hartmann Analytic) and incubated for 2 h at 37 °C. 100 ng in vitro transcribed tRNA product, synthesized acceptor stem of tRNA Gly-3 or mutations (RNA sequence was indicated in Fig. 4F, GenScript), 1 μg total RNA or 100 ng of *E. coli* tRNA was used per assay with 5 μM of protein. The 10 μl reactions were purified with Bio-Spin® 6 Columns (Bio-Rad), and mixed with 10 μl of RNA loading dye (95% formamide, 1 mM EDTA, 0.025% SDS, xylene cyanol and bromophenol blue), denatured at 90 °C and separated on 6% polyacrylamide-urea gels. The gel was vacuum dried at 80 °C, exposed to a phosphorimager screen and revealed by autoradiography using a Typhoon phosphorimager (GE Healthcare).

## In vitro transcription of tRNAs with homogeneous 3' ends

An optimized version of the hepatitis delta virus (HDV) ribozyme was used to generate homogeneous tRNA 3' ends as described[79]. Briefly, the DNA template T7-tRNA-HDV was amplified from plasmid pUC-57Kan-T7-tRNA-HDV (Supplementary Table S1), except T7-tRNA Gly-3-HDV was amplified by In-Fusion PCR. Labelled or unlabelled tRNAs were prepared by in vitro transcription of PCR templates using T7 RNA polymerase. The T7 RNA polymerase in vitro transcription reactions were performed in 25 μl total volume, with a 5 μl nucleotide mix of 2.5 mM ATP, 2.5 mM UTP, 2.5 mM GTP, 60 μM CTP (Promega, 10 mM stock) and 2–4 μl 10 mCi.ml$^{-1}$ of radiolabelled CTP [α–$^{32}$P], or with 5 μl nucleotide mix of 2.5 mM ATP, 2.5 mM UTP, 2.5 mM GTP, 2.5 mM CTP for unlabelled tRNA transcription. 50 to 100 ng of template were used per reaction with 1.5 μl rRNasin 40 U.ml$^{-1}$ (Promega), 5 μl 5x optimized transcription buffer (Promega), 2 μl T7 RNA polymerase (20 U.ml$^{-1}$) and 2.5 μl 100 mM DTT. Unincorporated nucleotides were removed by Micro Bio-Spin 6 columns (Bio-Rad) according to manufacturer's instructions. The transcripts were gel-purified on a denaturing 6% acrylamide gel and eluted in 0.3 M sodium acetate overnight at 20 °C. The supernatant was removed, ethanol precipitated and resuspended in 14 μl nuclease-free water. Radioactively labelled tRNAs carrying a 2′,3′ cyclic phosphate at the 3′ end were dephosphorylated using T4 polynucleotide kinase (NEB) in 100 mM Tris–HCl pH 6.5, 100 mM magnesium acetate and 5 mM β-ME in a final volume of 20 μl for 6 h at 37 °C. All assays were desalted by Micro Bio-Spin 6 columns (Bio-Rad).

## In vitro inhibition of MenT1 activity

MenA1 antitoxin activity was assayed using in vitro-transcribed tRNA Gly-3 as a substrate. For the co-incubation assay, MenT1 (5 μM) and increasing molar ratios of MenA1 were incubated with tRNA Gly-3 and 1 mM CTP in 10 μl reaction volumes containing 20 mM Tris–HCl pH 8.0 and 10 mM MgCl$_2$ for 4 h at 37 °C.

## MenT1 tRNA screening

The tRNA screening was performed using 100 ng *M. tuberculosis* tRNAs. The activity was tested in 20 mM Tris–HCl pH 8.0, 10 mM MgCl$_2$ and 1 mM CTP in 10 μl reaction volumes and incubated for 2 h at 37 °C. The transcripts were incubated with 5 μM MenT1 or with nuclease-free water as a control. The reaction was stopped with 10 μl RNA loading dye (95% formamide, 1 mM EDTA, 0.025% SDS, xylene cyanol and bromophenol blue), denatured at 90 °C and applied to 6% polyacrylamide-urea gels. The gel was vacuum dried at 80 °C and exposed to a phosphorimager screen, revealed by autoradiography using a Typhoon phosphorimager (GE Healthcare).

## tRNA library and sequencing

Primers used for the tRNA library are described in Supplementary Table S1. To obtain the MenT1 library, 5 µg total RNA of *M. smegmatis* with 1 mM CTP were incubated with water, 10 µg MenT1 WT or MenT1 D41A for 1 h at 37 °C. For the in vitro transcribed tRNA-seq, 15 ng of specific tRNA were incubated with 5 µM MenT1 WT or MenT1 D41A (1.3 µg) at 37 °C for 1 h. To obtain the MenT4 library, 5 µg total RNA of *M. smegmatis* with 1 mM GTP were incubated with water and 10 µg MenT4 WT for 2 h at 37 °C. Total RNA samples and single tRNA samples were isolated using trizol and ethanol precipitation, respectively. To remove m1A, m3C and m1G modification in the tRNAs, the total RNA samples were pretreated using the demethylase kit (Arraystar, cat#: AS-FS-004, Rockville, MD, USA), followed by trizol isolation. The 3p-v4 oligo was 5′ adenylated using 5′ DNA Adenylation Kit (E2610S, NEB) according to the protocol. To construct the library, RNA samples were ligated to the adenylated 3p-v4 adaptors, and reverse transcription was performed with ProtoScript II RT enzyme (NEB) using barcode primer (Supplementary Table S1). Finally, PCR amplification was performed with tRNAs oligoFor mix and A-PE-PCR10 (after 5 cycles, the programme was paused and add B_i7RPI1_CGTGAT or i7RPI7_GATCTG was added) using Q5 Polymerase Hot-Start (NEB). The library was sequenced by DNBSEQ-G400RS High-throughput Sequencing Set (PE100) in BGI Genomics (Hong Kong).

## RNA-Seq data processing

After a quality check, reads were demultiplexed to obtain a fastq file per experimental condition. For each experimental condition, the same procedure was applied: (i) mapping to a reference, (ii) PCR duplicates removal, (iii) quantification of read counts. Raw reads quality was checked with FastQC (http://www.bioinformatics.babraham.ac.uk/projects/fastqc). Further reads processing was performed with R software version 4.1.1 and BioConductor libraries for processing sequencing data obtained. R library ShortRead 1.50.0[80] was used to process fastq files. Rsubread 2.6.4[81] was used to map reads on the reference genome. Further filtering was performed to ensure reads contained the structure resulting from library preparation (Supplementary Table S1 for RNA-Seq read structure): reads start with a random nucleotide at position 1, a valid barcode at position 2–5, a recognition sequence resulting from ligation at position 6–24 with one mismatch allowed, random UMI sequence at position 25–39, agacat control sequence at position 40–45, and a nucleotide sequence at position 46–100 that corresponds to the reverse complement of the ligated 3′-end of tRNAs in the experiment. Reads corresponding to the different experimental conditions identified by the barcode were demultiplexed into fastq files for independent further analyses (mapping and quantification).

All the *M. smegmatis* mc$^2$155 (NC_008596.1) chromosomal tRNA gene sequences were extracted in a multifasta file. The sequence of genes rrfA, CspA, tmRNA were also added. Two additional sequences were added corresponding to *M. tuberculosis* H37RV (NC_000962.3) tmRNA and tRNA16-GlyTCC. The nucleotides CCA were concatenated at the end of each sequence. The reads were trimmed to only the RNA sequence from position 46 to 100 prior to the attempt to align them on reference sequences. Mapping on reference sequences was performed by using the library Rsubread 2.8.2 with parameters ensuring that the read maps without any mismatch or indel to a unique region: unique = TRUE, type = 'dna', maxMismatches = 0, indels = 0, ouput_format = 'BAM'.

For quantification, SAM files were processed with Rsamtools 2.8.0 (https://bioconductor.org/packages/Rsamtools) to count the number of reads mapping to the same region in reference sequences with the same 3′ unmapped sequences due to RNA 3′ end modifications. Reads were kept if the whole sequence mapped or if only the 3′ end did not map, which corresponds to nucleotides added by the toxin. In the alignments in the SAM file, this translates to a CIGAR code of only 'M' characters (indicating a match) possibly followed by 'S' characters (indicating no alignment to the reference sequence). The minimum length of the mapped region was set to 20 in length. PCR duplicates were removed using the UMI introduced in the library preparation for reads mapping exactly the same region (same beginning and end of the alignment) and having exactly the same unmapped 3′-end sequence. After these preprocessing steps, reads were regrouped by their 3′-end mapping position and unmapped 3′-end region sequence to quantify their abundance in terms of same tRNA 3′-end followed by the same nucleotides i.e. same post-transcriptional modifications. To remove noise, a tRNA with its post-transcriptional modifications was kept only if it was found at least 10 times in at least one of the experimental conditions.

Biological replicates and total reads of all the tRNA seq experiments are given in the Supplementary Datasheet 1 and uncropped gels in the Source data file.

## Statistics and reproducibility

No statistical method was used to predetermine sample size. No data were excluded from the analyses. The experiments were not randomized. The investigators were not blinded to allocation during experiments and outcome assessment.

## Reporting summary

Further information on research design is available in the Nature Portfolio Reporting Summary linked to this article.

## Data availability

The crystal structures of MenT$_1$ apo and the MenAT$_1$ complex have been deposited in the Protein Data Bank under accession numbers 8AN4 and 8AN5, respectively. All other data needed to evaluate the conclusions in the paper are present in the paper, and/or Supplementary Data. RNA-Seq data that support the findings of this study have been deposited in European Nucleotide Archive (ENA) at EMBL-EBI under accession number PRJEB62085. Source data are provided with this paper.

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

## Acknowledgements

This work was supported by a Springboard Award (SBF002\1104) from the Academy of Medical Sciences to B.U. and T.R.B.; the Centre National de la Recherche Scientifique, Université Paul Sabatier, Agence Nationale de la Recherche (ANR-19-CE12-0026) to O.N. and P.G.; the Programme d'Investissements d'Avenir (ANR-20-PAMR-0005) to O.N. and P.G.; the National Natural Science Foundation of China, (32000021) to X.X.; a scholarship from the China Scholarship Council (CSC) as part of a Joint International PhD programme with Toulouse University Paul Sabatier and a grant from the Fondation pour la Recherche Médicale (FDT202304016729) to X.H.; the Swiss National Science Foundation (CRSII3_160703) to P.G.; an Engineering and Physical Sciences Research Council Molecular Sciences for Medicine Centre for Doctoral Training studentship (EP/S022791/1) to T.J.A. We thank Yiming Cai and Léonora Poljak for their assistance and advice during the course of this study. We gratefully acknowledge Diamond Light Source for time on beamlines I24 and I04 under proposal MX24948.

## Author contributions

Analysed data: X.X., B.U., C.G., R.B., T.J.A., X.H., P.R., O.N., T.R.B. and P.G. Designed research: X.X., B.U., C.G., R.B., T.J.A., X.H., P.R., O.N., T.R.B. and P.G. Performed research: X.X., B.U., C.G., R.B., T.J.A., X.H. Wrote the paper: X.X, T.R.B. and P.G. with contributions from all the authors. Funding acquisition: X.X., O.N., T.R.B. and P.G. Supervised the study: T.R.B. and P.G.

## Competing interests

The authors declare no competing interests.
