## [Peer Review File · Nature Communications]

REVIEWER COMMENTS

Reviewer #1 (Remarks to the Author):

Xu et al present a biochemical and structural characterisation of the MenAT1 TA system of *Mycobacterium tuberculosis*. MenAT1 belongs to the MenAT family of modifying nucleotidyltransferase toxin-antitoxin systems. The study reveals a promiscuity in the specificity of modification by MenT1, and an unusual asymmetric tripartite toxin-antitoxin complex.

This is an illuminating, careful and well-written paper. It will be of interest to the bacterial toxin community, and potentially to the phage defence field also, given the role of MenT homologues in abortive infection (though I think evidence (or lack of) for/against phage defence as a possible function for MenT could be discussed more in the text).

My specific comments below are all relatively minor, but I do think annotated sequence alignment figures would be useful, and the issues with the FlaGs legend in the SI should be fixed.

- Figure 1A legend: I'm not sure why only three of the FlaGs-discovered conserved flanking genes get a place in the legend, and these are not even the most adjacent. For example I think there is a mistake and the turquoise gene is not the transcriptional regulator, but is a 'dormancy-associated translation inhibitor'. The original numbering from the FlaGs output with all of the annotations would have been useful here.
- Also in the FlaGs legend, 'GI number' should be 'NCBI protein accession'
- A FlaGs figure making tip: since the output order of 1A is the same as the tree (Fig S1), the two outputs can be stitched together to make one figure in Adobe Illustrator for example. Then the numbering can be reapplied in the SI fig, without cluttering the main text fig with numbers on genes.
- Also/alternatively perhaps a small inset box with a colour key in fig 1A for the most conserved flanking genes would be helpful so that the reader doesn't have to refer to the SI for the legend.
- End of Intro: 'The resulting unique TA complex differs from previously characterised TA complexes.' This is vague and not very informative. All structures are unique and differ from others in some way or another.
- Can anything be predicted about the biological function of MenAT1? Do the authors predict this is also an abortive infection system? Do the FlaGs results (a wider window would be useful here) suggest this system is located in a defence island? If not, are there any clues from the flanking genes - synteny is apparent at the genus level at least.

- It should be considered that IF MenAT1 is a phage defence system, its actual primary target may be a tRNA of the phage, explaining both a low specificity for host tRNAs and a lack of toxicity in *E. coli*. Since we don't know if this is a phage defence system at all, let alone which phage, this can't be tested yet. But is nevertheless worth mentioning.
- Maybe of interest to cite Dedrick et al 10.1038/nmicrobiol.2016.251 in the intro for a Mycobacterial (though in this case on a prophage) TA system that confers phage defence. The toxin, Gp29 (in the PhRel subfamily (not FaRel - see below!) of ToxSASs), is also a tRNA modifier (ref 51).
- Alignment figs showing a few representative seqs for both the toxin and the antitoxin would be useful for visualising conservation and structural info along the lengths of the proteins. These should be annotated with the limits of what is visible in the structures, the toxin active sites resides, the residues at the T:A interface, and the location of the introduced substitutions.
- After alignments are added, a sequence analysis section would be warranted, including the alignment tool (I recommend MAFFT with the L-ins-i strategy if there are <200 sequences) and info of FlaGs/webFlaGs analysis if not the default settings.
- Figure 5D-E: it would be helpful to see the MenT active site indicated on the structure (with a different colour for example), to orient relative to the antitoxin binding site.
- Page 9 line 194: 'most if not all' is quite vague and doesn't help the reader interpret the figure
- Discussion line 347: could differences in aa-tRNA synthetases also be behind the *E. coli*/ *M. tb* differences in toxicity?
- Discussion: "The FaRel subfamily of toxSAS toxins transfer a pyrophosphate moiety from ATP to the tRNA CCA end" This is incorrect. Among ToxSASs, FaRel synthesises toxic (pp)App, while it is PhRel, CapRel and FaRel2 that pyrophosphorylate tRNA.

Reviewer #2 (Remarks to the Author):

The manuscript by Xu et al. reports the high-resolution crystal structure of the mycobacterial toxin MenT1 alone as well as in complex with its antitoxin MenA1. MenT1 was recently identified by the authors as member of the DUF1814 nucleotidyltransferase toxins encoded by *Mycobacteria tuberculosis*. These toxins belong to the family of abortive infective proteins (AbiEii) of *Streptococci* belonging to the type IV family. In contrast to the latter, the authors provide strong evidence that MenT1 does belong to the classical type II TA family, in which the antitoxin MenA1 inhibits the toxic activity of MenT1 by formation of a heterotrimeric complex. In contrast to other members of the MenT inhibiting antitoxins, MenA1 is a short oligopeptide lacking any potential catalytic site or transcription regulatory domain. Ultimately, the authors show that MenT1 promiscuously modifies the 3' CCA tail of tRNA molecules by attaching single or multiple cytidine nucleotides thereby most likely preventing aminoacylation of the latter. Modification has been shown to occur in vitro on isolated tRNA as well as in coupled transcription/translation assays. Strikingly, in contrast to *M. tuberculosis* and *M. smegmatis*,

MenT1 did not exert any toxic phenotype when expressed in *E. coli*. The authors surmised that this might be due to the general tRNA abundance, as MenT1 only showed to interfere with translation efficiency *in vitro* once the concentration of tRNA was reduced.

The manuscript is written in a conclusive manner and the experimental data provided are convincing. However, there I have a few questions and remarks to the manuscript:

When isolating total RNA extracts from *M. smegmatis* and modifying the tRNA pool with MenT1, only 30 % of the tRNA molecules were modified with additional cytidine nucleotides at their 3' end. In contrast, when analyzing modification of *in vitro* transcribed tRNA, the authors report that almost all tRNA was modified by MenT1. Can the authors speculate why tRNA purified from *Msm* is apparently less modified than the *in vitro* transcribed counterparts? Was a portion of the isolated tRNA already aminoacylated? Can the authors isolate tRNA from *M. smegmatis* as well as *E. coli* and show to which extent the pool of tRNA molecules is modified *in vivo* after short expression of MenT1?

The authors make a rather strong claim that tRNA abundance, which they correlate with the number of different tRNA gene copies, is the cause that MenT1 exerts toxicity in mycobacteria but not in *E. coli* in the result section. Yet, in the discussion they state that tRNA maturation events etc. could cause this difference equally likely. In fact, when characterizing the MenT3 toxin, they recently showed that ectopic expression of RNase PH does desensitize mycobacteria as well as *E. coli* against MenT3 activity. What is the impact of RNase PH expression on MenT1 toxicity in mycobacteria? How can the authors exclude that this or a similar tRNA repair mechanism constitutively present in *E. coli* counteract MenT1 and thus not just the sheer number of tRNA genes is the cause of MenT1 failing to exert toxicity in *E. coli*? And further, what is the tRNA gene abundance in other organisms encoding a functional MenT1 for instance as listed in Figure 1 A.

Can the authors speculate from their structural data how MenT toxins establish specificity for nucleotides to be added to the CCA tail? As MenT toxins are structural similar to classical CCA adding enzyme, there is a plethora of structural reports on those enzymes for instance from the Steitz lab bound to different nucleotides. Does superposition with those help to identify critical residues for purine or pyrimidine specificity?

In addition to this more conceptual comments, the manuscript (including figures) could be improved when describing or showing results from X-ray diffraction experiments:

Starting lane 121 (similar also lane 230)

Should rather read something like ... we first solved (determined) the X-ray crystal structure of MenT1 alone to a resolution of 1.65 Å...

Figure 2: panel B: Mapping of the electrostatic potential onto the surface of MenT1. How was this calculated, did the authors use the APBS plugin module in pymol? What is the given contour level? Panel F: individual side chains which the authors apparently want to highlight are difficult to see.

Lane 599: Was indeed the entire MenA1:MenT1 heterotrimer used as search model or just one MenT1 protomer in MR?

Lane 603: Citation is missing

Reviewer #3 (Remarks to the Author):

Xu et al. characterized the substrate specificity and inhibition mechanism of MenT1, a toxin bearing nucleotidyltransferase activity. The authors uncovered that MenT3, a member of the uncharacterized toxin family, MenAT family, adds C on the end of tRNAs (PMID: 32923609). This paper characterized other members of MenAT family, MenAT1 and MenT4. Biochemical studies showed that MenT1 showed the same C-adding activity as MenT3, whereas MenT4 adds G on tRNAs. In contrast to MenT3, which preferentially targets class II tRNAs, MenT1 and MenT4 have broader substrate specificity. The authors also solve the crystal structure of the complex of MenT1 and MenA1, describing the detailed mechanism where the antitoxin MenA1 inhibits MenT1 activity by its binding. This inhibition mode is distinct from MenA3-mediated inhibition of MenT3 activity, highlighting the variability of toxin inactivation mechanisms even among the same toxin family. Most of the experiments are well performed, and the manuscript is well written. This paper will be of broad interest to researchers studying toxin-antitoxin and translation control mechanisms. However, the studies of substrate specificity are preliminary, and some results are inconsistent. Additional experiments will be necessary to establish the author's claims.

Major points

1. The authors use the word "promiscuity," but usage of this word is vague, so instead of using this word, explain specifically. For example, MenT1 appears not to distinguish between tRNAs, but it distinguishes tRNA from other RNAs. Also, the authors claimed that the MenT family has promiscuous substrate specificity. However, MenT3 has an apparent preference for class II tRNAs. A more explanation instead of using the word "promiscuity" clarifies the authors' claim.

2. The authors assessed the substrate specificity using in vitro assay combined with tRNA sequencing. However, whether the prepared total RNA fraction reflects the intracellular substrate condition is unclear. Since RNA extraction with the Trizol reagent remains most of charged aminoacylated tRNA, without deacylation, tRNA profile does not reflect the in vivo tRNA profiles. Moreover, in the cell, tRNA is in the cycle of aminoacylation and deacylation. Therefore, analysis of tRNA status in the cell, such as tRNA-seq of RNAs from MenT-expressing cells, is necessary for evaluating MenT's activities in the physiologically relevant condition.

3. Substrate specificity of MenT1 is a central topic of this manuscript. However, its mechanism is still unclear. Elucidating how MenT1 recognizes tRNA substrates will strengthen the authors' claim.

a. One mechanistic study showed that MenT1 requires A at the 3' end of the substrate in Fig. 4D. Does MenT1 recognize the sequence of CCA, or A at the 3' end is sufficient?

b. What are the other requirements? Acceptor stem sequence or clover leaf structure?

c. What is the minimum substrate for MenT1? Is minihelix with CCA end sufficient for MenT1 activity? This will uncover how MenT1 distinguish tRNAs from other RNAs.

d. What makes the preference for specific tRNAs?

e. Comparisons of substrate specificity between MenT proteins would be very valuable to strengthen the author's claim.

4. According to Fig. 4A and B, MenT1 prefers specific tRNAs, such as Arg-2, Glu1, Phe, but has almost no activities on several tRNAs, such as Trp, Gln-2. In addition, these patterns are not consistent with Fig3E. What is the cause of this inconsistency? It raises a concern that these experiments do not correctly examine the substrate specificity of MenT1.

Reviewer #1 (Remarks to the Author):

Xu et al present a biochemical and structural characterisation of the MenAT1 TA system of *Mycobacterium tuberculosis*. MenAT1 belongs to the MenAT family of modifying nucleotidyltransferase toxin-antitoxin systems. The study reveals a promiscuity in the specificity of modification by MenT1, and an unusual asymmetric tripartite toxin-antitoxin complex.

This is an illuminating, careful and well-written paper. It will be of interest to the bacterial toxin community, and potentially to the phage defence field also, given the role of MenT homologues in abortive infection (though I think evidence (or lack of) for/against phage defence as a possible function for MenT could be discussed more in the text).

My specific comments below are all relatively minor, but I do think annotated sequence alignment figures would be useful, and the issues with the FlaGs legend in the SI should be fixed.

- Figure 1A legend: I'm not sure why only three of the FlaGs-discovered conserved flanking genes get a place in the legend, and these are not even the most adjacent. For example I think there is a mistake and the turquoise gene is not the transcriptional regulator, but is a 'dormancy-associated translation inhibitor'. The original numbering from the FlaGs output with all of the annotations would have been useful here.

We thank the reviewer for the interesting suggestions. We have modified and improved the modified the FlaG figures (Fig. 1A and Fig. S1) as requested. Specifically, we have kept the flanking genes number in FlaGs output figure and given the protein details in supplementary datasheet 1.

- Also in the FlaGs legend, 'GI number' should be 'NCBI protein accession' Modified as requested, please see comment above.

- A FlaGs figure making tip: since the output order of 1A is the same as the tree (Fig S1), the two outputs can be stitched together to make one figure in Adobe Illustrator for example. Then the

numbering can be reapplied in the SI fig, without cluttering the main text fig with numbers on genes.

Thank you for the pointer. In this instance, we wanted to focus on the association of MenT1 and MenA1, rather than the phylogeny, and so opted to keep the phylogenetic output as Fig. S1.

- Also/alternatively perhaps a small inset box with a colour key in fig 1A for the most conserved flanking genes would be helpful so that the reader doesn't have to refer to the SI for the legend.

The outcome of our FlaGs search is that MenT1 does not appear to be associated with conserved region except perhaps for rv0078 and rv0077c that are present in the three related mycobacterial. In this case the only data available is that rv0078 could be a general regulator that also represses *menTAI* (Marie I. Samanovic, et al., 2018). As requested, we have now added the most conserved genes into the figure legend.

- End of Intro: 'The resulting unique TA complex differs from previously characterised TA complexes.' This is vague and not very informative. All structures are unique and differ from others in some way or another.

We agree with the reviewer and we have now deleted this sentence.

- Can anything be predicted about the biological function of MenAT1? Do the authors predict this is also an abortive infection system? Do the FlaGs results (a wider window would be useful here) suggest this system is located in a defence island? If not, are there any clues from the flanking genes - synteny is apparent at the genus level at least.

Based on the FlaG analysis and the Mtb genome, we can only say that *menTAI* is located in one of the proposed large genomic islands of Mtb. Accordingly, we have now added a reference to this work (Becq et al. Mol. Biol. Evol. 24(8):1861–1871. 2007 Contribution of Horizontally Acquired Genomic Islands to the Evolution of the Tubercle Bacilli - Table1): results section line 108. Although it is annotated as AbiEii in FlaGs, (see updated supplementary Datasheet 1) it is not possible to predict its function in phage defence or else.

- It should be considered that IF MenAT1 is a phage defence system, its actual primary target may be a tRNA of the phage, explaining both a low specificity for host tRNAs and a lack of toxicity in E. coli. Since we don't know if this is a phage defence system at all, let alone which phage, this can't be tested yet. But is nevertheless worth mentioning.

Although there are no native Mtb phage isolated, many mycobacterial phages do indeed have tRNA in their genomes and we thus fully agree that if MenT1 and 4 could target specific phage tRNA, which is a very interesting point.

The following sentence has been added in the discussion line 387:

“One attractive possibility is that toxins with low specificity towards the host tRNA, *i.e.*, MenT1 and MenT4, would specifically target and prevent aminoacylation of phage encoded tRNA^{8,46}, thus acting as phage defence systems.”

- Maybe of interest to cite Dedrick et al 10.1038/nmicrobiol.2016.251 in the intro for a Mycobacterial (though in this case on a prophage) TA system that confers phage defence. The toxin, Gp29 (in the PhRel subfamily (not FaRel - see below!) of ToxSASs), is also a tRNA modifier (ref 51).

We agree with the reviewer, Dedrick and colleagues and the possible targeting of phage tRNA has now been discussed (see above).

- Alignment figs showing a few representative seqs for both the toxin and the antitoxin would be useful for visualising conservation and structural info along the lengths of the proteins. These should be annotated with the limits of what is visible in the structures, the toxin active sites resides, the residues at the T:A interface, and the location of the introduced substitutions.

We thank the reviewer for this suggestion. The figure has been generated and added as Fig. S2, and the following line has been added at line 110:

“Sequence alignments of the identified MenT1 and MenA1 homologues demonstrates conservation of structural elements and putative active site residues (Fig. S2).”

- After alignments are added, a sequence analysis section would be warranted, including the alignment tool (I recommend MAFFT with the L-ins-i strategy if there are <200 sequences) and info of FlaGs/webFlaGs analysis if not the default settings.

Sequence analysis has been performed and the results are provided as Fig. S2. Details of how these alignments were performed have been added into the Materials and Methods section, lines 444-450.

- Figure 5D-E: it would be helpful to see the MenT active site indicated on the structure (with a different colour for example), to orient relative to the antitoxin binding site.

The figure panels have been altered to show the locations of the four active site residues for each protomer, which we agree helps the reader understand the relative positions compared to MenA1. Thank you for the suggestion.

- Page 9 line 194: 'most if not all' is quite vague and doesn't help the reader interpret the figure

This has been altered to “... almost all tRNA, 42 out of the 45 in this case, could be labelled ...” (line 203).

- Discussion line 347: could differences in aa-tRNA synthetases also be behind the E. coli/ M. tb differences in toxicity?

This is an interesting possibility. We have now added aminoacyl-tRNA synthetases in the proposed differences in the discussion section, now line 364.

- Discussion: "The FaRel subfamily of toxSAS toxins transfer a pyrophosphate moiety from ATP to the tRNA CCA end" This is incorrect. Among ToxSASs, FaRel synthesises toxic (pp)App, while it is PhRel, CapRel and FaRel2 that pyrophosphorylate tRNA.

Sincere apologies for this oversight. The manuscript has been changed to read, "The ToxSAS toxins PhRel, CapRel and FaRel2 transfer a pyrophosphate moiety from ATP to the tRNA CCA end".

Reviewer #2 (Remarks to the Author):

The manuscript by Xu et al. reports the high-resolution crystal structure of the mycobacterial toxin MenT1 alone as well as in complex with its antitoxin MenA1. MenT1 was recently identified by the authors as member of the DUF1814 nucleotidyltransferase toxins encoded by *Mycobacteria tuberculosis*. These toxins belong to the family of abortive infective proteins (AbiEii) of *Streptococci* belonging to the type IV family. In contrast to the latter, the authors provide strong evidence that MenT1 does belong to the classical type II TA family, in which the antitoxin MenA1 inhibits the toxic activity of MenT1 by formation of a heterotrimeric complex. In contrast to other members of the MenT inhibiting antitoxins, MenA1 is a short oligopeptide lacking any potential catalytic site or transcription regulatory domain. Ultimately, the authors show that MenT1 promiscuously modifies the 3' CCA tail of tRNA molecules by attaching single or multiple cytidine nucleotides thereby most likely preventing aminoacylation of the latter. Modification has been shown to occur in vitro on isolated tRNA as well as in coupled transcription/translation assays. Strikingly, in contrast to *M. tuberculosis* and *M. smegmatis*, MenT1 did not exert any toxic phenotype when expressed in *E. coli*. The authors surmised that this might be due to the general tRNA abundance, as MenT1 only showed to interfere with translation efficiency in vitro once the concentration of tRNA was reduced.

The manuscript is written in a conclusive manner and the experimental data provided are convincing. However, there I have a few questions and remarks to the manuscript:

We thank the reviewer for the positive comments and constructive criticism.

-When isolating total RNA extracts from *M. smegmatis* and modifying the tRNA pool with MenT1, only 30 % of the tRNA molecules were modified with additional cytidine nucleotides at their 3' end. In contrast, when analyzing modification of in vitro transcribed tRNA, the authors report that almost all tRNA was modified by MenT1. Can the authors speculate why tRNA purified from *Msm* is apparently less modified than the in vitro transcribed counterparts? Was a portion of the isolated tRNA already aminoacylated? Can the authors isolate tRNA from *M. smegmatis* as well as *E. coli* and show to which extent the pool of tRNA molecules is modified in vivo after short expression of MenT1?

Our tRNA seq data show a semi-quantitative view of the bulk of tRNA modified by MenT1 (i.e., about 30% of the total tRNA reads obtained) and of each separate tRNA that could be identified by this method (Fig 3D and E). In contrast, the *in vitro* tRNA assay with single unlabeled tRNA purified following T7 transcription in the presence of labeled CTP (to label the CMP modification) only shows if a tRNA can be modified or not by MenT1 (Fig. 4A), providing no clue about the modified fraction of a single tRNA. Therefore, by “almost all tRNA were modified by MenT1” we intended to state that our screen revealed that almost each tRNA individually could be modified by MenT1, which merely confirms the tRNA seq experiment. Whether such approaches can be performed *in vivo* with short expression of MenT1 appears to be very challenging and will need further extensive work.

We have now modified the text line 199 to clarify this point:

“To further confirm that MenT1 can modify most of the tRNA, each of the 45 tRNAs of *M. tuberculosis* were individually transcribed *in vitro* using T7 RNA polymerase and screened for tRNA modification by MenT1 in the presence of [α - 32 P]-CTP. Such a non-quantitative screen, which only detects 3' modifications of tRNA, confirmed that almost all tRNA, 42 out of the 45 in this case, could be labelled with CMP in the presence of MenT1 (Fig. 4A).”

Yet, we fully agree with the reviewer that *in vivo* purified tRNA could be a better target than *in vitro* purified single tRNA. This is exemplified by the tRNAseq data of our Gly-3 model tRNA, which seems to be less efficiently modified when purified *in vitro* than from the *in vivo* total tRNA sample (compare 13% in Fig S5 vs 40% in Fig 3D). This suggests that *in vivo* purified mature tRNAs, which are subjected to many modifications, could be better MenT substrates. Although very speculative at this stage, this is an interesting aspect of specificity that will be one of the next challenges in the field.

We have added the following sentence to highlight such an interesting possibility in the text line 367:

“For example, it is not known whether native tRNA modifications that occur during their maturation could influence their recognition and modification by MenT1. The fact that our model Gly-3 tRNA was more heavily modified by MenT1 when purified from *in vivo* cell extracts (~40%) than from *in vitro* synthesis (~13%) suggests that mature tRNA could be more efficiently targeted by MenT1. ”

-The authors make a rather strong claim that tRNA abundance, which they correlate with the number of different tRNA gene copies, is the cause that MenT1 exerts toxicity in mycobacteria but not in *E. coli* in the result section. Yet, in the discussion they state that tRNA maturation events etc. could cause this difference equally likely. In fact, when characterizing the MenT3 toxin, they recently showed that ectopic expression of RNase PH does desensitize mycobacteria as well as *E. coli* against MenT3 activity. What is the impact of RNase PH expression on MenT1 toxicity in mycobacteria? How can the authors exclude that this or a similar tRNA repair mechanism constitutively present in *E. coli* counteract MenT1 and thus not just the sheer number of tRNA genes is the cause of MenT1 failing to exert toxicity in *E. coli*? And further, what is the

tRNA gene abundance in other organisms encoding a functional MenT1 for instance as listed in Figure 1 A.

We fully agree with the reviewer that endogenous tRNA repair mechanism in *E. coli* might counteract MenT1 more efficiently; the discussion about tRNA concentration in *E. coli* is just a hypothesis among others. The only evidence that we have so far is that decreasing the concentration of *E. coli* tRNA *in vitro* favors translation inhibition by MenT1. We are very cautious at this stage because too little is known about how the cell would cope with these toxins.

Concerning RNase PH, we could only detect partial rescue of MenT toxicity in *E. coli* but never in mycobacteria, under all the conditions tested so far. Yet, it is clear that co-expression systems in mycobacteria are less productive than for *E. coli*, which could be an further issue to consider. Whilst *M. tuberculosis* RNase PH might be critical the fact that (in contrast with *E. coli*) most Mtb tRNA do not have CCA ends (30 out of 45) suggests that robust CCA adding activity might also be required.

The sentence modified in response to Reviewer 1 integrates such comment, line 363:

“In addition, differences in tRNA modification, aminoacyl-tRNA synthetases, repair mechanisms (*e.g.*, RNase PH) and maturation (*e.g.*, CCA-adding enzyme), or the lack of specific partners in *E. coli*, could also play a role.”

-Can the authors speculate from their structural data how MenT toxins establish specificity for nucleotides to be added to the CCA tail? As MenT toxins are structural similar to classical CCA adding enzyme, there is a plethora of structural reports on those enzymes for instance from the Steitz lab bound to different nucleotides. Does superposition with those help to identify critical residues for purine or pyrimidine specificity?

We are very grateful to the reviewer for this interesting suggestion. We performed the structural superpositions for two relevant enzymes, and they were indeed insightful. We didn't observe evidence to help explain nucleotide specificity but we certainly gained insight towards nucleotide binding modes, and how a tRNA is likely presented to the active site. The superpositions have been shown as new Fig. S4, and an additional paragraph has been added into the results on page 8, line 154:

“Structural superposition of MenT1 with ANT2 (PDB 4XJE) demonstrated a potential binding mode for nucleotide substrates, with the triphosphate co-ordinated through divalent metal ions supported by conserved MenT1 residues D41 and D43 (**Fig. S4A**). Structural superposition of MenT1 with a CCA-adding enzyme from *A. fulgidus* (PDB 3OVA) shows not only a similar potential nucleotide binding mode (**Fig. S4B**), but also the potential positioning for a tRNA substrate, wrapping over the MenT1 surface and presenting the 3' end at the active site (**Fig. S4C**).”

-In addition to this more conceptual comments, the manuscript (including figures) could be improved when describing or showing results from X-ray diffraction experiments:

Starting lane 121 (similar also lane 230)

Should rather read something like ... we first solved (determined) the X-ray crystal structure of MenT1 alone to a resolution of 1.65 Å...

Changed as requested.

-Figure 2: panel B: Mapping of the electrostatic potential onto the surface of MenT1. How was this calculated, did the authors use the APBS plugin module in pymol? What is the given contour level? Panel F: individual side chains which the authors apparently want to highlight are difficult to see.

We have added details of the electrostatics into the figure legend for 2B. We have increased the size of the relevant section of the image for 2F, and altered the labelling so that it is less cluttered.

-Lane 599: Was indeed the entire MenA1:MenT1 heterotrimer used as search model or just one MenT1 protomer in MR?

Apologies for the oversight, this has been clarified; just one MenT1 protomer was used.

-Lane 603: Citation is missing

Apologies, this has been added.

Reviewer #3 (Remarks to the Author):

Xu et al. characterized the substrate specificity and inhibition mechanism of MenT1, a toxin bearing nucleotidyltransferase activity. The authors uncovered that MenT3, a member of the uncharacterized toxin family, MenAT family, adds C on the end of tRNAs (PMID: 32923609). This paper characterized other members of MenAT family, MenAT1 and MenT4. Biochemical studies showed that MenT1 showed the same C-adding activity as MenT3, whereas MenT4 adds G on tRNAs. In contrast to MenT3, which preferentially targets class II tRNAs, MenT1 and MenT4 have broader substrate specificity. The authors also solve the crystal structure of the complex of MenT1 and MenA1, describing the detailed mechanism where the antitoxin MenA1 inhibits MenT1 activity by its binding. This inhibition mode is distinct from MenA3-mediated inhibition of MenT3 activity, highlighting the variability of toxin inactivation mechanisms even among the same toxin family. Most of the experiments are well performed, and the manuscript is well written. This paper will be of broad interest to researchers studying toxin-antitoxin and translation control mechanisms. However, the studies of substrate specificity are preliminary, and some results are inconsistent. Additional experiments will be necessary to establish the author's claims.

Major points

1. The authors use the word “promiscuity,” but usage of this word is vague, so instead of using this word, explain specifically. For example, MenT1 appears not to distinguish between tRNAs, but it distinguishes tRNA from other RNAs. Also, the authors claimed that the MenT family has promiscuous substrate specificity. However, MenT3 has an apparent preference for class II tRNAs. A more explanation instead of using the word “promiscuity” clarifies the authors’ claim.

We fully agree with the reviewer. We have now deleted all the references to “promiscuity” and explained more specifically the preference of MenT1 throughout the manuscript. We have also modified the manuscript title.

2. The authors assessed the substrate specificity using *in vitro* assay combined with tRNA sequencing. However, whether the prepared total RNA fraction reflects the intracellular substrate condition is unclear. Since RNA extraction with the Trizol reagent remains most of charged aminoacylated tRNA, without deacylation, tRNA profile does not reflect the *in vivo* tRNA profiles. Moreover, in the cell, tRNA is in the cycle of aminoacylation and deacylation. Therefore, analysis of tRNA status in the cell, such as tRNA-seq of RNAs from MenT-expressing cells, is necessary for evaluating MenT’s activities in the physiologically relevant condition.

We agree with the reviewer that our study specifically describes the activity of MenT1, only *in vitro*. We are fully aware that a thorough analysis of MenT toxins *in vivo*, especially in the native host *M. tuberculosis* will be a necessary step towards the comprehension of these toxins. Yet, we sincerely believe that this work is a significant new challenge that is beyond the scope of our current study, which aims to focus on the *in vitro* activity and structures of MenT1 and its MenA1 cognate antitoxin. At present, many technical issues need to be solved, especially in *M. tuberculosis* cells expressing the toxins, before we can rigorously address these important questions *in vivo*.

3. Substrate specificity of MenT1 is a central topic of this manuscript. However, its mechanism is still unclear. Elucidating how MenT1 recognizes tRNA substrates will strengthen the authors’ claim.

a. One mechanistic study showed that MenT1 requires A at the 3’ end of the substrate in Fig. 4D. Does MenT1 recognizes the sequence of CCA, or A at the 3’ end is sufficient?

In order to address such questions experimentally, we have constructed six additional mutations in the CC positions of the CCA end, i.e., CAA, CTA, CGA, ACA, TCA, GCA of the Gly-3 model tRNA, purified and labeled them, and tested them *in vitro* for modification by MenT1 under the same condition as Fig. 4C and D. These data, presented in the new Fig. 4E show that, within the context of Gly-3 modification, mutations at these positions had little (CTA and GCA) or no effect (CAA, CGA, ACA and TCA) for MenT1, which is in sharp contrast with any modification in the last adenosine position (Fig. 4D).

In addition to new Fig. 4E, the following text has been added at line 221:

“In addition, mutations of the CC positions of the CCA end of the Gly-3 tRNA (*i.e.*, CAA, CTA, CGA, ACA, TCA and GCA ends) exhibited little (CTA and GCA) or no detectable effect (CAA, CGA, ACA and TCA) on modification by MenT1(Fig. 4E), which is in sharp contrast with the inhibition effect of mutation in the last adenosine (Fig. 4D).”

- b. What are the other requirements? Acceptor stem sequence or clover leaf structure?
- c. What is the minimum substrate for MenT1? Is minihelix with CCA end sufficient for MenT1 activity? This will uncover how MenT1 distinguish tRNAs from other RNAs.

We have experimentally addressed comments b and c together by preparing a minihelix acceptor stem of Gly-3 tRNA and several of its variants with successive deletions of the UCCA-3' end, and tested them for MenT1 modification in the presence of labeled CTP (new Fig. 4F). Interestingly, the data show that the Gly-3 tRNA minihelix acceptor stem containing the wild-type CCA end is still modified by MenT1, while as expected, deletion of the last adenosine was sufficient to inhibit modification.

In addition to new Fig. 4F, the following text has been added line 225:

“In order to investigate whether the tRNA acceptor stem alone was sufficient to be recognized and modified by MenT1, a Gly-3 tRNA acceptor stem construct and several of its variants with successive deletions of the UCCA-3' end were tested for modification by MenT1 in the presence of labeled CTP (Fig. 4F). In this case, we found that the Gly-3 tRNA acceptor stem containing the wild-type CCA end could still be modified by MenT1. Consistent with the data obtained with the full length Gly-3 tRNA, deletion of the last adenosine was sufficient to inhibit modification by MenT1.”

- d. What makes the preference for specific tRNAs?

The fact that several known tRNA targeting toxins target a specific tRNA instead of many of them suggests that it might be advantageous to have one essential target. In the case of MenT3, which appears to be specific towards serine tRNA of *Mtb in vitro*, we can hypothesize that some structural elements of these tRNA or perhaps even their aa-tRNA synthetase could play a role in such specificity. As we discussed from line 392 to line 402, specific tRNA sequence (for example, CreT RNA toxin sequesters tRNA-Arg; VapC4 cleaves a single site within the anticodon sequence of tRNA-Cys) and the amino acid residue type of charged tRNA (for example, *Escherichia coli* ItaT toxin acetylates aminoacyl-tRNAs charged with hydrophobic residues) could determine the preference for specific tRNAs. Yet, in the case of MenT1 or MenT4, which are less toxic than MenT3 and have low or no apparent preference for specific tRNA, we could expect that difference *in vivo* in tRNA charging or tRNA modifications (see our reply to reviewer 2, manuscript line 364) could be critical for these toxins to be active. In agreement with reviewer 1, one could expect that the TA systems could specifically target tRNA from bacteriophages, as a defence mechanism. This has now been added in the discussion line 388.

We have now tried to address this more clearly in the discussion in addition to the other reviewers' comments:

Line 382

“Whether specific sequences or structural elements of serine tRNA would contribute to such preference by MenT3 remains unknown. To date, there is no structural information suggesting why MenT3 but not MenT1 or MenT4, shows such a preference in *in vitro*. Finally, MenT4 activity also showed no apparent preference for specific tRNA and a less robust toxicity than MenT3²⁵, thus highlighting a continuum of target selection throughout the MenT toxin family. One attractive possibility is that toxins with low specificity towards the host tRNA, *i.e.*, MenT1 and MenT4, would specifically target and prevent aminoacylation of phage encoded tRNA^{8,46}, thus acting as phage defence systems.”

e. Comparisons of substrate specificity between MenT proteins would be very valuable to strengthen the author’s claim.

In response to Reviewer 1 we performed structural superpositions to allow analysis of substrate binding by MenT1. We didn’t observe evidence to help explain tRNA or nucleotide specificity but we certainly gained insight towards nucleotide binding modes, and how a tRNA is likely presented to the active site. The superpositions have been shown as new Fig. S4, and an additional paragraph has been added into the results on page 8, line 154.

4. According to Fig. 4A and B, MenT1 prefers specific tRNAs, such as Arg-2, Glu1, Phe, but has almost no activities on several tRNAs, such as Trp, Gln-2. In addition, these patterns are not consistent with Fig3E. What is the cause of this inconsistency? It raises a concern that these experiments do not correctly examine the substrate specificity of MenT1.

The data from Fig. 4A represents a large screening of all Mtb tRNA T7-transcribed *in vitro* and directly incubated with MenT1 and [α -32P]-CTP. In this case, we can only detect if a labeled cytidine is added to a specific tRNA in the presence of MenT1, without any indication about (i) the number of cytidines that are added to a specific tRNA and (ii) the proportion of a given tRNA that has been modified. In addition, T7 transcribed products 3’end can be highly heterogenous, which could also affect modification by MenT1. In this screen, it is thus possible that we miss some weaker tRNA modification or that some tRNA purified under these conditions cannot be efficiently recognized by MenT1. Yet, this screen, which allows the rapid screening of all tRNA *in vitro* largely confirm the data obtained with the total tRNA isolated *in vivo* and analyzed by tRNAseq (Figs. 3D,E).

For these reasons, we next selected five representative tRNAs from the tRNAseq and the radioactive screen (including one, Leu-3, that is not detected in the Fig 4A screen but modified following tRNA seq in Fig. 3E) to perform a more thorough analysis using single [α -32P]-labeled tRNA purified after HDV cleavage (Fig.4B). This supports our conclusion that MenT1 does not appear to be very selective.

REVIEWERS' COMMENTS

Reviewer #1 (Remarks to the Author):

Thanks to the authors for addressing all the points I raised. I have no further comments.

Reviewer #2 (Remarks to the Author):

The authors have carefully addressed the reviewer's comment in the revised manuscript and convincingly clarified potential concerns in their rebuttal letter. New and exciting results have been included and I would like to congratulate the authors to this exciting manuscript.

Reviewer #3 (Remarks to the Author):

The authors adequately addressed the concerns raised by this reviewer.